# Aglycosylated antibody-producing mice for aglycosylated antibody-lectin coupled immunoassay for the quantification of tumor markers (ALIQUAT)

Nan-Ee Lee[1,2,6], Sun Hee Kim[1,6], Dae-Yeul Yu[1], Eui-Jeon Woo  [2,3], Myung-Il Kim[4], Gi-Sang Seong[4], Sun Min Lee[5], Jeong-Heon Ko[1,2 ✉] & Yong-Sam Kim [1,2 ✉]

Targeting aberrant glycoforms has been validated for in vitro cancer diagnostic development, and several assays are currently in routine clinical use. Because N-glycans in Fc region of antibodies show cross-reactivity with various lectins, high-quality aglycosylated antibodies are exceptionally important for immunoassay platform-based quantitative measurements. Previously, aglycosylated antibody acquisition relied on incomplete, uneconomical and onerous enzymatic and chemical methods. Here, we edited four murine immunoglobulin G genes using adenine base-editing and homology-directed recombination (HDR)-mediated gene editing methods to generate aglycosylated antibody-producing mice. Resulting aglycosylated antibodies showed required analytical performances without compromised protein stability. Thus, this aglycosylated monoclonal antibody-lectin coupled immunoassay for the quantification of tumour markers (ALIQUAT) method can provide a robust, versatile and accessible immunoassay platform to quantify specific glycoforms in precision cancer diagnostics. Moreover, the engineered mice can be used as a host to produce various aglycosylated antibodies in a convenient and robust fashion, thereby expanding in vitro diagnostic development opportunities that utilize glycoforms as a disease-specific biomarkers.

[1] Genome Editing Research Center, KRIBB, Daejeon 34141, Republic of Korea. [2] Department of Bio-molecular Science, KRIBB School of Bioscience, Korea University of Science and Technology (UST), Daejeon 34113, Republic of Korea. [3] Disease Target Structure Research Center, KRIBB, Daejeon 34141, Republic of Korea. [4] Bioneer, Daejeon 34302, Republic of Korea. [5] Department of Laboratory Medicine, Pusan National University Yangsan Hospital and School of Medicine, Gyeongnam 50612, Republic of Korea. [6] These authors contributed equally: Nan-Ee Lee, Sun Hee Kim. ✉email: jhko@kribb.re.kr; omsys1@kribb.re.kr

Aberrant protein glycosylation is closely associated with various pathological conditions including cancer[1], thereby justifying efforts to use a specific glycoform of a protein as a disease biomarker. In fact, glycoproteins account for a large portion of Food and Drug Administration (FDA)-approved cancer biomarkers. Furthermore, an altered glycan structure or a specific glycoform is considered as an analyte for in vitro cancer diagnostics[2,3]. Best-characterized biomarkers include serum alpha-fetoprotein (AFP), which is an oncofetal protein that is frequently overexpressed in hepatocellular carcinoma (HCC)[3]. However, increased serum levels of AFP are also observed under non-HCC conditions, including liver inflammation and cirrhosis, pregnancy, and so on[4,5]. The high false-positive rate of AFP has made it unfeasible to use the serum levels of AFP for screening or early diagnosis[6]. In contrast, the *Lens culinaris* agglutinin (LCA)-reactive, core-fucosylated form of AFP (AFP-L3) is a more specific indicator than total AFP for HCC, and an AFP-L3/total AFP ratio ≥10% is highly specific for early HCC or non-seminomatous germ cell tumors[7]. An automated assay measuring the AFP-L3 percentage, known as a μ-TAS AFP-L3 analyzer, received FDA approval for risk assessment of HCC development. More recently, an immunoassay to measure a specific glycoform of Mac-2-binding protein glycan isomer was developed to monitor patients with liver fibrosis and cirrhosis[8]. The immunoassay can also be applied to predict the development of HCC in hepatitis B patients treated with nucleot(s)ide analogs[9]. Rigorous efforts to relate glyco-biomarkers to disease status are underway and recent developments of high-throughput technologies in genomics, proteomics, and metabolomics are expected to facilitate such efforts.

The μ-TAS AFP-L3 analyzer is an automated system that was originally developed for AFP tests. The assay is relatively expensive and, furthermore, applications of biomarkers other than AFP require substantial efforts for assay developments,

which restricts its routine use in clinical settings. In contrast, enzyme-linked immunoadsorbent assay (ELISA) or chemiluminescent immunoassay (CLIA) platforms provide a simple, versatile, and robust assay platform with reduced costs, and the development of new tests can be easily achieved without the need for complicated instrumentation. As such, a specific glycoform could be measured in an ELISA or CLIA platform by using a capture antibody and a glycoform-specific probe, such as a lectin as schematized in Fig. 1a. However, there is a fundamental challenge when applying those platforms to measure a specific glycoform. Most of the capture antibodies used in CLIA or ELISA platforms belong to the immunoglobulin G (IgG) class, which is unexceptionally N-glycosylated in the constant region of the heavy chains[10]. Thus, there are unwanted but unavoidable interactions between antibodies and lectins, thereby making such immunoassays fundamentally unfeasible. Several efforts have been made to overcome this problem, including deglycosylation using PNGase-F[11], enzymatic cleavage of antibodies using pepsin[12] or papain[13], and chemical modifications of glycans[14]. There are, however, several drawbacks for such approaches: (1) it is hard to achieve complete enzymatic or chemical reactions and it requires extra efforts to quality-control deglycosylation; (2) an additional purification step is required and a significant loss in the antibodies is unavoidable; (3) these laborious efforts should be individually and redundantly made for each in vitro test.

To resolve these issues, we genome-engineered mice by editing N-glycosylation sites (asparagine-coding codons) of IgG into non-asparagine-coding sequences. We found that Fc-aglycosylated antibodies were normally produced from genome-engineered mice and that such aglycosylated antibodies retained high stability equivalent to that of conventional N-glycosylated antibodies. With this finding, we suggest a method, termed *a*glycosylated antibody-*l*ectin-coupled *i*mmunoassay for the *qua*ntification of tumor markers (ALIQUAT), in which aglycosylated antibodies produced

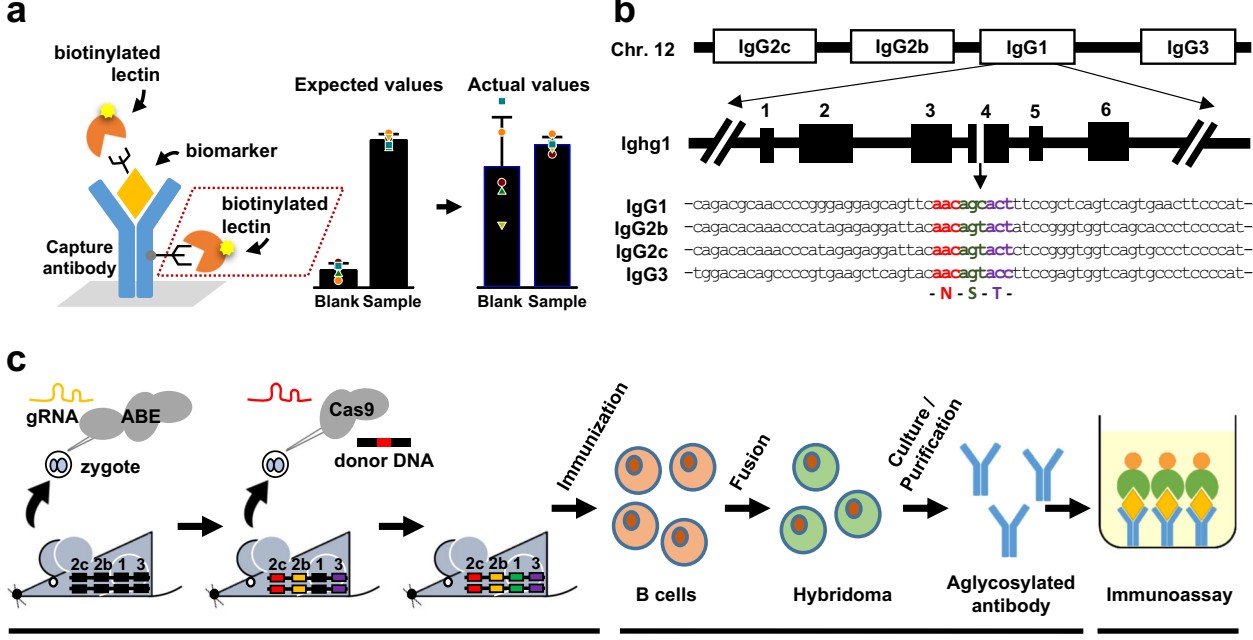

**Fig. 1 Overall scheme for the generation of mice producing aglycosylated IgGs. a** Obstacles of glycosylated antibodies in the quantification of glycoforms as illustrated for an ELISA platform. A glycoform-specific lectin binds to glycosylated capture antibodies independently of analytes, causing saturated blank values and high standard errors. **b** The conserved nucleotide sequence encoding an N-glycosylation motif (N–S–T) sequence in exon 4 of CH2 of *Ighg1*, *Ighg2b*, *Ighg2c*, and *Ighg3* genes. **c** Overall scheme for genome engineering of mice (phase 1), production of aglycosylated antibodies (phase 2), and the development of ALIQUAT (phase 3).

from the engineered mice are used to capture glycoprotein bio-markers and a lectin probe for a specific glycoform. ALIQUAT, when applied to AFP-L3 tests, yielded assay results substantially identical to μ-TAS AFP-L3 analysis and, furthermore, is expected to serve as a universal and versatile platform for various glyco-biomarker assays. In addition, the genome-engineered mice can be readily used as a host for the generation of aglycosylated anti-bodies against a wide range of biomarkers.

## Results

**Experimental scheme**. IgGs are N-linked glycoproteins that show heterogeneity in the glycan structure, as glycoproteins usually do. This molecular nature has restricted the use of IgGs for the detection of glycoprotein biomarkers on a routinely used immunoassay platform. As illustrated with ELISA in Fig. 1a, a glycoform-specific lectin can be bound to the capture antibody even in the absence of analytes. Moreover, the glycan structure is not easy to predict and is subject to batch-to-batch variations, thereby producing uncontrollable, saturated blank values. To address this problem, we adopted a straightforward strategy different from the previous strategies[11–14]; once mice are genome engineered, so that they produce an aglycosylated antibody, a specific glycoform can be measured in a simple immunoassay platform through the coupled use of a glycoform-specific lectin and aglycosylated anti-bodies, referred to as ALIQUAT herein. Commonly used mouse strains, including C57BL/6, have four IgG subclasses, *IgG1*, *IgG2b*, *IgG2c*, and *IgG3*[15]. Those genes are linked on chromosome 12 and invariably have conserved sequences encoding the N-glycosylation consensus sequence (Asn–Ser–Thr) in the exon 4 (Fig. 1b). To generate aglycosylated antibody-producing mice, we aimed to correct asparagine into non-asparagine encoding sequences.

The workflow of this study consists of three experimental phases: phase 1, genome engineering of mice; phase 2, production of aglycosylated antibodies; and phase 3, establishment of the ALIQUAT method. First, genome engineering was performed using the C57BL/6 strain, which is readily handled for microinjection and has well-established genomic information. Because the four subclass genes are linked in a single chromo-some, genome engineering procedures were carried out so that a series of mutations were stacked in the single chromosome. In the second phase, monoclonal aglycosylated antibodies were produced from the engineered mice. In the third phase, we suggested that ALIQUAT provides a valid platform for measuring glyco-biomarkers by comparing ALIQUAT with μ-Tas AFP-L3 analysis for measurement of AFP/%AFP-L3 as a case study.

**Genome engineering to generate aglycosylated IgG-producing mice**. Although several methods have been suggested to improve the homology-directed recombination (HDR) efficiency[16], it still remains low compared to non-homologous end-joining (NHEJ)-based knockout efficiency. Recently, base-editing systems have been developed in which "C–G to T–A" and "A–T to G–C" conversions can be made with no double-stranded DNA break-age, but rather with high efficiency by cytosine[17,18] and adenine base editors[19], respectively. Sequence analysis of IgG genes of C57BL/6 mice revealed that *IgG2b*, *IgG2c*, and *IgG3* share editable adenines at an N-glycosylation sequence (AAC) within a base editing window (Fig. 1b). The conversion of either one or both of adenines to guanine creates non-asparagine-coding sequences, thereby resulting in aglycosylation of IgGs. As an initial approach, we injected ABE7.10 mRNA and a mixture of guide RNAs tar-geting IgG2g, IgG2c, and IgG3 into zygotes of C57BL/6 mice. Genotype screening was performed for pups born ($n = 24$) by the amplification refractory mutation system (ARMS), in which pri-mers were designed to generate an additional PCR product of a

lower size when A to G conversion was made as indicated with an arrow (Fig. 2a–c and Supplementary Fig. 1). The assay results indicated that highly efficient A to G conversions were achieved for the three genes targeted; the conversion rates (conversion of at least one locus) were 79.1% (19/24) for *IgG2c*, 87.5% (21/24) for *IgG2b*, and 62.5% (15/24) for *IgG3*. Comparison of the screening results identified 15 pups that appeared to show simultaneous conversions at the three genes, nine of which were subjected to Sanger sequencing analysis (Supplementary Fig. 2). Surprisingly, all nine pups were identified to carry mutations at the three gene loci if chimeric mutations were included. Among them, one pup (#11) was selected for the next round of genome engineering at the IgG1 locus, because it carried more biallelic "A to G" con-versions in the *IgG2b*, *IgG2c*, and *IgG3* genes than other pups (Fig. 2a–c). The pup had a homozygous mutation at *IgG2b* and chimeric mutations at *IgG2c* and *IgG3* (Supplementary Fig. 3). We performed several rounds of crossing and the finally gener-ated pups carried one of the mutant amino acid motifs, D-S-T, D-G-T, G-S-T, and G-G-T in IgG2a, IgG2c, and IgG3 loci. We selected a D-S-T mutant pup as an intermediate founder for the following reasons: the N-glycosylation asparagine residue is located at the exposed loop between two strands of the immu-noglobulin β-sandwich fold, and aspartic acid exhibits an iden-tical geometry to that of asparagine. Moreover, the charged side chain is a preferred choice to glycine to avoid potential geome-trical instability of the region (Supplementary Fig. 4). This intermediate founder was bred and used for the engineering of the IgG1 gene.

The target site of IgG1 contains no canonical (NGG), but only the NG PAM sequence at both strands of the target. Under this situation, we had two technical options. One was to use an xCas9- or Cas-NG-adenine base editor that retains gene targeting activity for an NG PAM sequence[20,21], and the other is to perform an HDR-mediated gene correction. We attempted both approaches, but could not generate pups with intended conversions using xCas9-ABEs, presumably due to target-dependent low efficiency. For HDR-mediated gene correction, the target sequence was selected among 5′-upstream of the N-glycosylation site (Fig. 2d). The donor DNA was designed as PCR amplicons to carry the "Asp to Glu" mutation at the glycosylation site so that the mutation can create a SacI reporter sequence as well as a PAM mutation. IgG1 engineering was performed by injecting a ribonucleoprotein complex (RNP) comprising SpCas9 proteins and synthetic dual guide RNAs, SCR7[22] and donor DNA into zygotes. A restriction enzyme digestion assay using SacI revealed that 4 out 13 pups born showed a cleavage product, but three of the four pups were found to carry additional mutations at an unintended site. Only one pup (#34) carried an intended, heterozygous mutation. The engineered pup was also subjected to several rounds of cross until a homozygous IgG1 mutation was achieved. As expected, all these mutations showed a germline transmission and the founder with stacked aglycosylation mutations at *IgG1*, *IgG2b*, *IgG2c*, and *IgG3* was successfully established. The mutations at the four alleles were again confirmed by Sanger sequencing. Additional detailed information on the overall genome engineering procedures is compiled in Table 1.

Although we could obtain a founder mouse with stacked IgG mutations, there still remained a concern that unintended off-target activity may affect antibody production or any other unwanted physiological perturbations. Thus, we investigated genome-wide mutations by comparing genome sequences of wild-type C57BL/6 and the engineered mouse (pup#34). We adopted the Iofreq, strelka, and mutect2 methods for the comparative analysis[23], and identified 286 single nucleotide variations (SNVs). However, there were no indels throughout the genome that were commonly identified by the three methods (Fig. 2e, Supplementary Data 1). Next, we

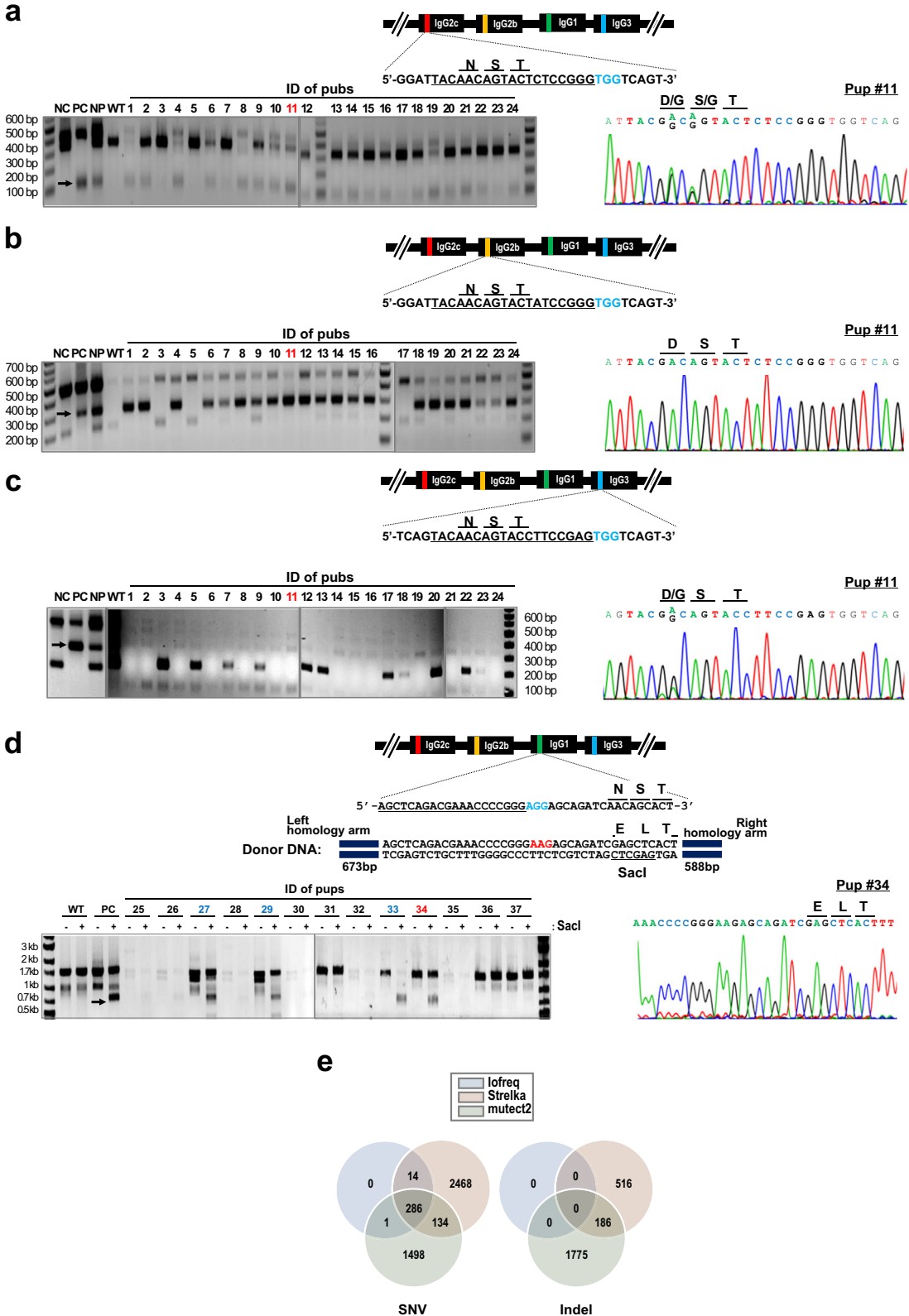

**Fig. 2 Genome engineering of mice at the N-glycosylation sites of IgG genes. a–c** Mutation of *IgG2c* (**a**), *IgG2b* (**b**), and *IgG3* (**c**) gene using the SpCas9-ABE7.10 base-editing system. ARMS assays and the Sanger sequencing results of *Ighg2c* (**a**), *Ighg2b* (**b**), and *Ighg3* (**c**) mutations are shown for a selected pup (#11). The pup had biallelic A to G conversions for the three genes, creating non-asparagine-coding sequences. For the ARMS assay, please refer to Supplementary Fig. 1. **d** Mutation of *IgG1* gene via HDR-mediated gene correction. Only one pup (#34) was born with an intended monoallelic mutation following injection of an RNP, SCR7, and donor DNA into zygotes. The donor DNA was designed to carry a SacI reporter sequence as well as a PAM mutation. **e** The genome-wide SNVs and indels were investigated for engineered pup#34 using the lofreq, Strelka, and mutect2 methods. SNVs and indels that were identically identified by the three methods were considered off-target mutations. NC negative control, PC positive control, NP negative–positive control mixture, WT wild type.

**Table 1 Summary of genome engineering to generate aglycosylated antibody-producing mice.**

| Subclass | Method | Effector protein | Effector/gRNA | Amino acid modification | Donor type | Supplement | No. of eggs | | | Pups born | Mutants (monoallelic/biallelic) |
|---|---|---|---|---|---|---|---|---|---|---|---|
| | | | | | | | Collected | Injected | Transferred | | |
| IgG1 | HDR | SpCas9 | Protein/RNA | NST→ELT | PCR amplicons | SCR7 | 211 | 196 | 32 | 13 | 1/0 |
| IgG2b IgG2c IgG3 | Adenine base editing | ABE7.10 | mRNA/RNA | NST→DST | N/A | — | 405 | 317 | 112 | 24 | 10/11 12/7 5/10 |

searched out SNVs that result in changes in the amino acid sequence in exons or may affect gene expression in the regulatory regions and consensus sequences for splicing. The sequence analysis identified 12 mutation sites corresponding to the mentioned criteria, seven of which belonged to the on-target IgG genes. Mutations at 5 sites of Prrc2b, Henmt1, BC080695, Deftb35, and Kcnip1 were considered off-target mutations (Table 2). To clear any potentially harmful mutations, we carried out further rounds of back-crossing and selections. Finally, we established a founder that carries wild-type sequences at the off-target sites (Supplementary Fig. 5).

**Production of aglycosylated monoclonal antibodies in the engineered mice**. The N-glycosylation of IgG is highly conserved among higher eukaryotic organisms[24]. Although there are reports that IgG N-glycosylation confers stability[25], or regulates the immune response[26], which evolutionary pressure has made higher organisms adopt N-glycosylation of IgG is still elusive. This question may evoke a concern that our genome-engineered mice may show defective IgG production. To test this possibility, IgG profiling was conducted using an immunoglobulin isotyping kit. The engineered mice displayed an identical profile to that of a wild-type C57BL/6 mouse (Fig. 3a), whereas mice with knockout mutations at the IgG2b and IgG3 genes showed defects in the production of the corresponding subclasses (Supplementary Fig. 6). This result confirmed that the engineered mice retain the capability to produce every subclass of IgGs.

Next, we produced an aglycosylated antibody from the engineered mice. We opted to use human AFP (hAFP) as a model antigen, which is one of the best-characterized tumor markers for HCC[27]. hAFP emulsified with an equal amount of adjuvants was injected intracutaneously into the hind footpads of 6-week-old engineered mice (n = 5) five times at an interval of 1 week each. Sera were collected at each immunization step and used for direct ELISA tests against hAFP, revealing an increased reactivity over time (Fig. 3b). After immunizations were completed, popliteal lymph node cells were collected and fused with FO myeloma cells. Hybridoma cells were allowed to form colonies on hypoxanthine-aminopterin-thymidine (HAT)-containing plates. In total, we obtained 666 hybridoma clones, among which 92 clones (13.8%) secreted an antibody that was highly reactive to hAFP (Supplementary Fig. 7). Because anti-hAFP antibodies may have cross-reactivity for human serum albumin (HSA) depending on epitopes[28], we also checked reactivity for HSA and reactive clones were excluded. In addition, there were clones that showed decreased antibody productions over time. These selections enabled narrowing down to three clones, designated 1E5, 2A2, and 3A5. The clones secreted a monoclonal antibody of the IgG1 subclass and kappa light chains (Fig. 3c). The monoclonal antibody produced from the 1E5 clone was purified on a protein G-affinity column and subjected to peptide sequencing on a mass spectrometer. The MS spectra confirmed the intended, non-N-glycosylation sequence E-L-T (Fig. 3d). There were no post-translational modifications in the mutated regions that were newly formed, such as O-linked glycosylation. To further confirm that the produced antibody is completely aglycosylated, lectin blot analysis using concanavalin A (Con-A) and immunoblot analysis using anti-mouse IgG antibodies were performed. In contrast to a commercially purchased antibody, the heavy chains of the antibody (1E5) had no trace of N-glycans, as indicated by an arrow (Fig. 3e). Recent studies revealed that 15% IgGs have Fab glycans and their patterns show distinct structural features depending on health conditions[19,29,30]. Accordingly, we further investigated whether 1E5 clone is actually aglycosylated, but it showed no trace of N-glycosylation (Supplementary Fig. 8).

 5

**Table 2 Single nucleotide variations that may affect gene expression or cause changes in the amino acid sequences.**

| Chr. no. | Location | Gene | Position | Codon | | Amino acid | | Target |
|---|---|---|---|---|---|---|---|---|
| | | | | Ref. | #34 | Ref. | #34 | |
| Chr. 2 | 32,216,742 | Prrc2b | Exon | A | C | K | N | Off-target |
| Chr. 3 | 108,941,384 | Henmt1 | 5′-UTR | G | A | | | Off-target |
| Chr. 4 | 143,572,327 | BC080695 | Exon | G | A | G | D | Off-target |
| Chr. 8 | 21,940,715 | Defb35 | Exon | G | A | R | Q | Off-target |
| Chr. 11 | 33,992,821 | Kcnip1 | Exon | T | C | K | E | Off-target |
| Chr. 12 | 113,287,956 | Ighg2c | Exon | T | C | N | D | On-target |
| Chr. 12 | 113,306,956 | Ighg2b | Exon | T | C | N | D | On-target |
| Chr. 12 | 113,329,548 | Ighg1 | Exon | G | T | S | L | On-target |
| Chr. 12 | 113,329,549 | Ighg1 | Exon | A | C | | | |
| Chr. 12 | 113,329,550 | Ighg1 | Exon | C | G | N | E | On-target |
| Chr. 12 | 113,329,552 | Ighg1 | Exon | A | G | | | |
| Chr. 12 | 113,360,232 | Ighg2c | Exon | T | C | N | D | On-target |

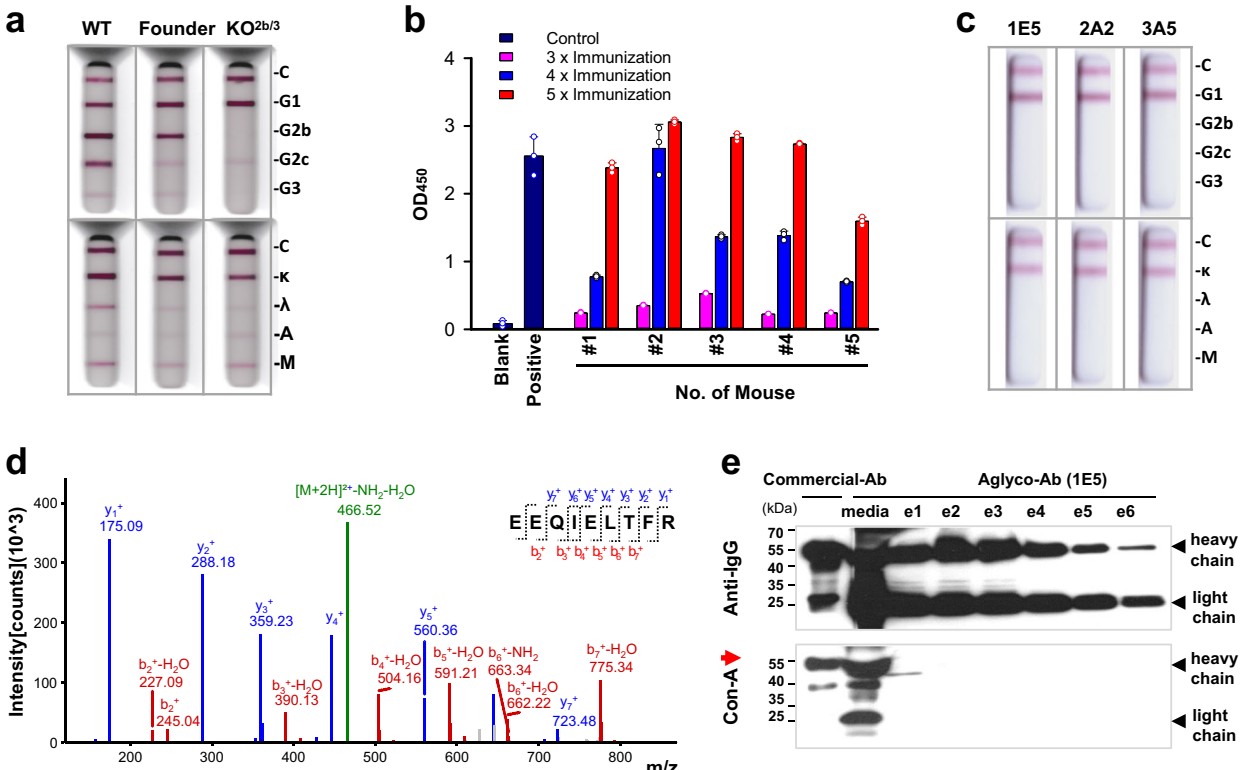

**Fig. 3 Production of aglycosylated monoclonal antibodies raised against hAFP from the engineered mice. a** Immunoglobulin profiles in sera of the engineered mice. The engineered mice normally produced IgGs of all subclasses. **b** Reactivity of the sera to hAFP collected from mice ($n = 5$) following immunizations with hAFP proteins. Values are means ± standard deviation, hereafter. **c** Identification of a subclass of aglycosylated monoclonal antibodies produced in the engineered mice. The three clones belonged to IgG1, with kappa light chains. **d** An MS-based peptide sequencing result of a monoclonal antibody, 1E5. The glycosylation consensus sequence (N–S–T) was correctly mutated to the intended E–L–T sequence. **e** Confirmation of the aglycosylated status of the 1E5 antibody. The monoclonal antibody was purified on a protein G-affinity column, and each elution fraction was run on an SDS-PAGE gel. Heavy and light chains were identified by immunoblot analysis using an anti-mouse IgG antibody. Aglycosylation status was assessed by lectin blot analysis using concanavalin A.

**Feasibility test of lectin-coupled ELISA.** To establish the ALI-QUAT system, we attempted to test the feasibility of the produced aglycosylated antibody in an ELISA platform. To this end, it was critical to obtain hAFP standards that consisted of pure AFP-L1 and AFP-L3. HEK293-T cells highly express α-(1,6)-fucosyltransferase (FUT8), which catalyses the transfer of fucose to the core N-glycan[31]. To produce an L3-null AFP standard, the FUT8 gene was ablated by CRISPR-Cas9. There are three splicing variants for the FUT8 gene, and exons 9 and 11 belong to

glycosyltransferase family 23[32], which is associated with catalytic activity and is shared by all the variants. The two exons were separately targeted, and we obtained each $FUT8^{-/-}$ mutant clone with a frame-shifted non-sense mutation (Fig. 4a). To investigate functional loss by FUT8 gene ablation, lectin binding was tested using a lectin from the mushroom *Pholiota squarrosa* (PhoSL), which shows a specific binding property for core-fucosylated N-glycans[33]. Immunofluorescence results indicated that $FUT8^{-/-}$ HEK293-T cells were resistant to PhoSL binding (Fig. 4b).

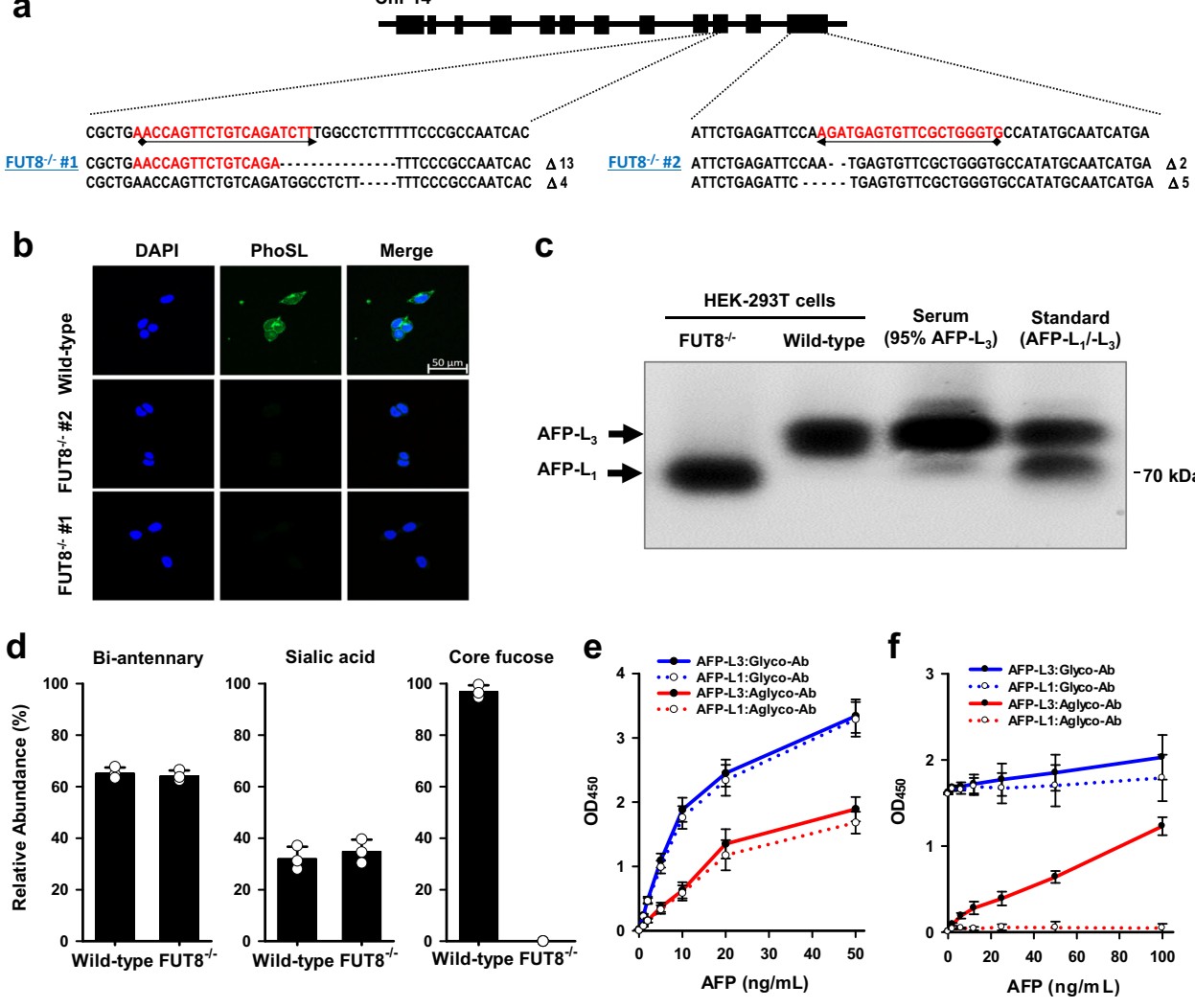

**Fig. 4 Feasibility tests of a PhoSL-coupled ELISA for the measurement of total AFP and AFP-L3. a** Ablation of the *FUT8* gene in HEK293-T cells using CRISPR-Cas9. Exons 9 and 11 were separately targeted, and each *FUT8⁻/⁻* clone that showed a frame-shifted non-sense mutation was established. **b** Confirmation of *FUT8* ablation through immunofluorescence using PhoSL. Both *FUT8⁻/⁻* HEK293-T mutant cells were resistant to PhoSL binding. A 50-μm scale bar is presented. **c** Production of AFP-L1 and AFP-L3 samples for the feasibility test of aglycosylated antibodies. AFP-L1 and AFP-L3 of high purity were produced from wild-type and *FUT8⁻/⁻* HEK293-T cells, respectively. The two glycoforms showed a different mobility on LCA-embedded agarose gels. Each AFP was identified by immunoblot using an anti-hAFP antibody. **d** Confirmation of the purity of AFP-L1 and AFP-L3 by mass analysis. **e**, **f** Quantitative measurement of total AFP (**e**) and AFP-L3 (**f**). A commercially available glycosylated antibody and an aglycosylated antibody (1E5) were used as capture antibodies. PhoSL was used for probing AFP-L3 (**f**). Values are means ± standard deviations for triplicate measurements.

hAFP was overexpressed in wild-type or *FUT8⁻/⁻* HEK293-T cells, and the conditioned media were collected for lectin-affinity electrophoresis. Proteins were run on agarose gels containing LCA, on which AFP-L3 and AFP-L1 show different mobility[34]. The different glycoforms of AFP were visualized by immunoblotting using an anti-hAFP antibody (Fig. 4c). AFP-L3 comprised the majority of the AFP glycoforms obtained from wild-type cells, whereas *FUT8⁻/⁻* cells did not produce AFP-L3. Mass analysis also confirmed the absence of the AFP-L3 form in *FUT8⁻/⁻* cells (Fig. 4d and Supplementary Fig. 9), and μ-Tas AFP-L3 analysis found that the %AFP-L3 produced in the wild-type cells was 99.5%. In addition, the absence of AFP-L3 in *FUT8⁻/⁻* cells was confirmed by μ-Tas AFP-L3 analysis. Using these recombinant proteins, a standard curve was produced by employing both a canonical sandwich ELISA (Fig. 4e) and a lectin-ELISA using PhoSL. For the canonical sandwich ELISA, an aglycosylated antibody (1E5) or a commercial antibody were used as a capture antibody. A detection antibody pair was selected from

commercially available antibodies, which are known to be a matched pair with the commercial capture antibody. The standard curve indicated that, *albeit* with a lower sensitivity than the validated commercial antibody, the aglycosylated antibody showed good linearity in the range of 0–20 ng/ml AFP. The feasibility of our aglycosylated antibody was further demonstrated by AFP-L3 tests (Fig. 4f). The commercial antibody showed saturated blank values, and increased concentrations of AFP-L3 did not reflect a linear increase in the optical density, resulting in only high standard errors. This pattern was similarly observed for both L3-positive and -negative AFP samples. However, the lectin-ELISA test using PhoSL and our aglycosylated antibody produced a standard curve with good linearity for L3-positive AFP samples. No interference was observed in the range of 0–100 ng/ml, and L-fucose inhibited the AFP-L3-specific binding of the lectin (Supplementary Fig. 10), thereby confirming that ALIQUAT method may provide a valid analytical platform for the assay of specific glycoforms.

**ALIQUAT produces results that are substantially identical to those of μ-TAS AFP-L3 analysis.** The μ-TAS AFP-L3 test is an in vitro diagnostic test that basically utilizes a different electrophoretic mobility of AFP glycoforms in a capillary tube. The high sensitivity and robustness of the analyzer and clinical utility of the AFP-L3 assay have allowed the test to gain FDA approval for the assessment of HCC development risk[35]. Despite the analytical validity and clinical utility, μ-TAS AFP-L3 analysis has several limitations in clinical use; typically, it is a high-cost assay, which has limited its routine clinical use for cancer screening. Moreover, further analytical developments are required to test glycoform biomarkers other than AFP-L3 under the μ-TAS system. For these reasons, the ALIQUAT method could replace or be used interchangeably with the μ-TAS AFP-L3 test and further be a versatile and universal platform for the detection of a wide range of disease-specific glycoforms. To explore this possibility, we attempted to prove analytical validity using clinical samples on the ELISA-based ALIQUAT platform.

First, the linearity of the calibration curves was investigated. Pure AFP-L1 and AFP-L3 samples were spiked into a normal serum that contained a negligible amount of AFP (below 1 ng/ml) so that a total AFP concentration was kept constant at 100 ng/ml, and AFP-L3 / AFP-L1 ratios were variable from 0:100 to 100:0%. The calibration curves showed excellent linearity, with an $R^2$ of 0.989–0.997 in the AFP-L3 range of 0–100 ng/ml (Fig. 5a). Next, the reliability of ALIQUAT was assessed by investigating intra- and inter-rater reliability using an intraclass

correlation coefficient (ICC)[36]. AFP-L3 was repeatedly measured from a total of 50 serum samples by a single rater. Two independent measures were statistically significant ($p = 0.00078$), and an excellent intra-rater reliability was observed, with an ICC measure of 0.991. Three independent raters were involved to investigate inter-rater reliability, and the ICC estimate was 0.967. Because the lower estimates of 95% confidence intervals (CIs) were >0.80 in both intra- and inter-reliability tests, we concluded that the ALIQUAT method showed excellent methodological reliability (Table 3). Next, we performed method agreement analysis. Forty-one randomly selected clinical samples were analyzed in terms of AFP/AFP-L3 levels using the μ-TAS AFP-L3 and ALIQUAT method. The comparative quantitative measurement indicated substantially overlapped AFP vs. AFP-L3 plots (Fig. 5b). To gauge the level of difference in the two methods, we performed Bland–Altman analysis[37]. The Bland–Altman analysis is a graphical method to compare two measurements techniques, one of which can be a reference or "gold standard" method. Usually, horizontal lines are drawn at the mean difference, and at the limits of agreement, which is defined by an average difference ± 1.96 times of standard deviation of the difference. The Bland–Altman plot revealed that there appeared to be no systemic errors for ALIQUAT methods and the difference was random over the tested range (Fig. 5c). Although compared to μ-TAS AFP-L3 test, ALIQUAT underestimated the %AFP-L3, the difference was marginal, with the mean difference being −0.83% in the measurement of %

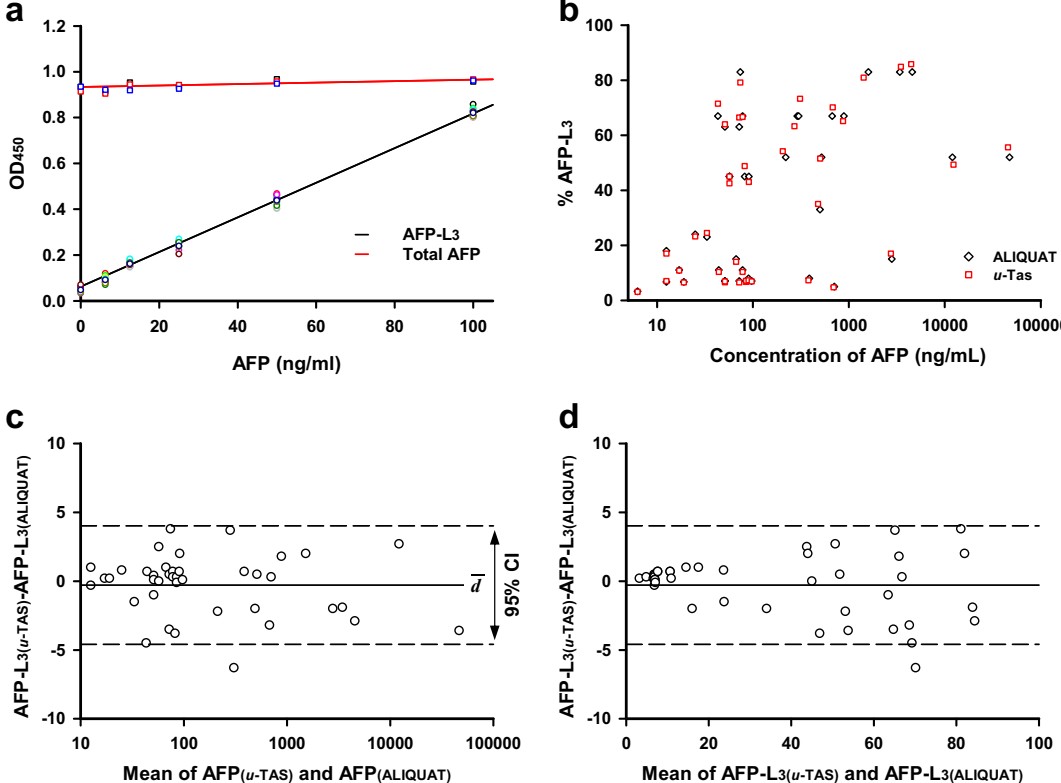

**Fig. 5 Analytical validation of the ALIQUAT method.** **a** Standard curves for the quantification of total AFP and AFP-L3 in the ALIQUAT platform. Different concentrations of an AFP-L1/AFP-L3 mixture and AFP-L3 were spiked into normal sera and standard curves were constructed. Total AFP and AFP-L3 were measured by canonical sandwich ELISA and ALIQUAT, respectively; $n = 9$. **b** Total AFP/AFP-L3 was measured by both the ALIQUAT and μ-TAS AFP-L3 methods. Sera ($n = 41$) were collected from normal volunteers and patients with hepatitis, cirrhosis and HCC and split into equal volumes, each of which were used to measure total AFP and AFP-L3 through the ALIQUAT method and μ-TAS AFP-L3 analysis. The measured values were plotted as the total AFP level on the $X$-axis and % AFP-L3 on the $Y$-axis. **c, d** Bland–Altman plot for data from Table 3. The difference in the measure of AFP-L3 was plotted versus the mean of total AFP (**c**) and AFP-L3 (**d**). The solid and the broken lines indicate the mean difference ($d$) between the ALIQUAT and μ-TAS AFP-L3 analysis and 95% CI, respectively.

**Table 3 Assessment of the reliability of ALIQUAT using an intraclass correlation coefficient.**

| | Intra-rater reliability ($n = 50$) | | | | Inter-rater reliability ($n = 40$) | | | |
|---|---|---|---|---|---|---|---|---|
| | Test 1 (ng/ml) | Test 2 (ng/ml) | *p* Value | ICC (95% CI) | Tester 1 (ng/ml) | Tester 2 (ng/ml) | Tester 3 (ng/ml) | ICC (95% CI) |
| Value | 49.9 ± 6.0 | 49.1 ± 4.2 | <0.001 | 0.991 (0.985–0.995) | 47.0 ± 34.0 | 45.9 ± 36.7 | 46.9 ± 35.1 | 0.967 (0.946–0.982) |

AFP-L3. The 95% CI of agreement limits was $-4.59 \leq CI \leq +4.02$. Although there are still no established criteria on the acceptable CIs in the AFP/AFP-L3 tests, we suggest that the difference may be acceptable considering the fact that AFP-L3 tests focus on the cut-off values for clinical decision (usually, ≥10% AFP-L3, ≥8 ng/ml AFP). In fact, the difference was marginal in the critical range of ~0–10% AFP-L3, especially with <2.0% (Fig. 5d), where a slight difference in %AFP-L3 could affect the diagnostic decisions on the risk of HCC. From these results, we concluded that the ALIQUAT method provides a reliable analytical platform for AFP/AFP-L3 tests and could be used interchangeably with the µ-TAS AFP-L3 test. In addition to analytical validity, a further large validation study needs to be conducted to establish a cut-off value of AFP-L3 for the ALIQUAT method to secure clinical validity.

**Preserved protein stability of aglycosylated antibodies.** Protein glycosylation is known to confer protein stability in vivo and in vitro[38], which may make one wonder if aglycosylated antibodies show loss of protein stability during storage and tests. Thus, we investigated the stability of antibodies under various conditions and compared the aglycosylated antibodies with commercially available glycosylated antibodies and the deglycosylated form produced by PNGase-F treatment according to the manufacturer's instructions. First, we placed those different forms of antibodies at 4 and 37 °C up to 14 days in phosphate-buffered saline (PBS) buffer. The deglycosylated antibody showed two different degradation fragments (Fig. 6a). One of the fragments, with a molecular mass of ~35 kDa, is thought to arise from the PNGase-F treatment per se because the fragment was observed from the early incubation time. The abundance of another fragment band observed at a molecular mass of ~50 kDa increased with time. However, the original glycosylated form did not show any degradation products under the investigated conditions. Similarly, we could not detect any trace of degradation products for aglycosylated antibodies on the gels. These results indicate that some harsh conditions for the deglycosylation reaction account for the loss of stability, rather than the absence of the glycan itself.

Antibodies may be exposed to changes in pH, according to various reaction conditions or during purification. We compared the stabilities of the different glycoforms of antibodies under different pH conditions. After the antibodies were incubated in buffers of pH 3.0, 7.0, and 10.0 at 37 °C for up to 14 days, the antibodies were used as a capture antibody in the measurements of total hAFP. The optical density values were used as a surrogate for assessing stability. The glycosylated and aglycosylated antibodies showed an almost identical pattern (Fig. 6b). Both antibodies showed a highly stable avidity up to 14 days under neutral (pH 7.0) and high resistance under basic (pH 10.0) conditions, while their stability decreased in a time-dependent manner at pH 3.0. In contrast, the deglycosylated form showed a significant loss in integrity even in neutral pH. Almost half of the avidity was lost after incubation at pH 7.0 for 14 days, and the loss was more prominent for incubations at pH 10.0. Notably, the deglycosylated form showed a complete loss of avidity after

incubation at pH 3.0 for 4 days. The integrity of antibodies was also tested under oxidative conditions. Both glycosylated and aglycosylated antibodies showed resistance against such conditions, which was in contrast to the physical properties of the deglycosylated form (Fig. 6c). The aglycosylated antibody showed a stability loss of ≤ 10% when incubated in PBS buffer containing up to 3.0% $H_2O_2$ for 1 h. However, a deglycosylated antibody lost almost half of its avidity. Taken together, these results led us to conclude that aglycosylated antibodies did not show compromised protein stability, compared to that of glycosylated IgGs.

## Discussion

Glycan-free antibody can be achieved through various technical approaches. Traditionally, researchers have purchased commercially available antibodies and treated them in-house with PNGase-F in a defined buffer system to eradicate N-glycans from the purchased antibodies. The most prominent problem lies in the fact that antibody N-glycans are somehow concealed in the Fc region and resistant to digestion by PNGase-F[39]. For this reason, the buffer systems usually contain detergents, such as NP-40. Our results (Fig. 6) show that the reaction condition is not favorable to the integrity of antibodies. Moreover, it has not been guaranteed that the deglycosylation reaction can be reached to perfection: the degree of deglycosylation is time-dependent, and a long incubation time may be needed to obtain fully deglycosylated antibodies. However, the longer time antibodies are exposed to such conditions, the more compromised the integrity of antibodies is. In addition, it is often difficult to remove detergent even after deglycosylation, which may severely affect antibody activity depending on the types of measurements. The issue of an incomplete reaction is identically applied to another enzymatic approach: pepsin or papain can be used to cut near the hinge region of constant chains. There have been attempts to use cleaved Fab fragments as a capture antibody. However, extra effort is needed to investigate the fully deglycosylated status. In addition, the orientation of the surface-coated antibodies is often limited for binding to the corresponding antigens[40]. Whether it is PNGase-F, pepsin, or papain, the used enzymes should be separated from the deglycosylated products after the reaction, which requires extra effort and often results in a significant loss in antibodies, apart from the deglycosylation efficiency. Chemical modifications were attempted as an alternative method to obtain deglycosylated antibodies[11–14], but they usually depends on reducing sugars and thus are restricted to several sugar moieties, such as sialic acids. Genome engineering can be performed for hybridoma cells to achieve aglycosylation, but it should be redundantly conducted for each hybridoma clone. Similarly, a single-chain Fv can be used as an alternative for aglycosylated antibody, but the fusion of $V_H$ and $V_L$ should be done on an individual antibody basis. In contrast, the generation of our genome-engineered mice is expected to provide a radical solution to the aforementioned restrictions. Researchers or commercial vendors have to only use engineered mice to produce aglycosylated antibodies without in vitro enzymatic reactions and purification. In addition to simplicity, our mice provide a means to obtain aglycosylated antibodies in a more time- and cost-saving

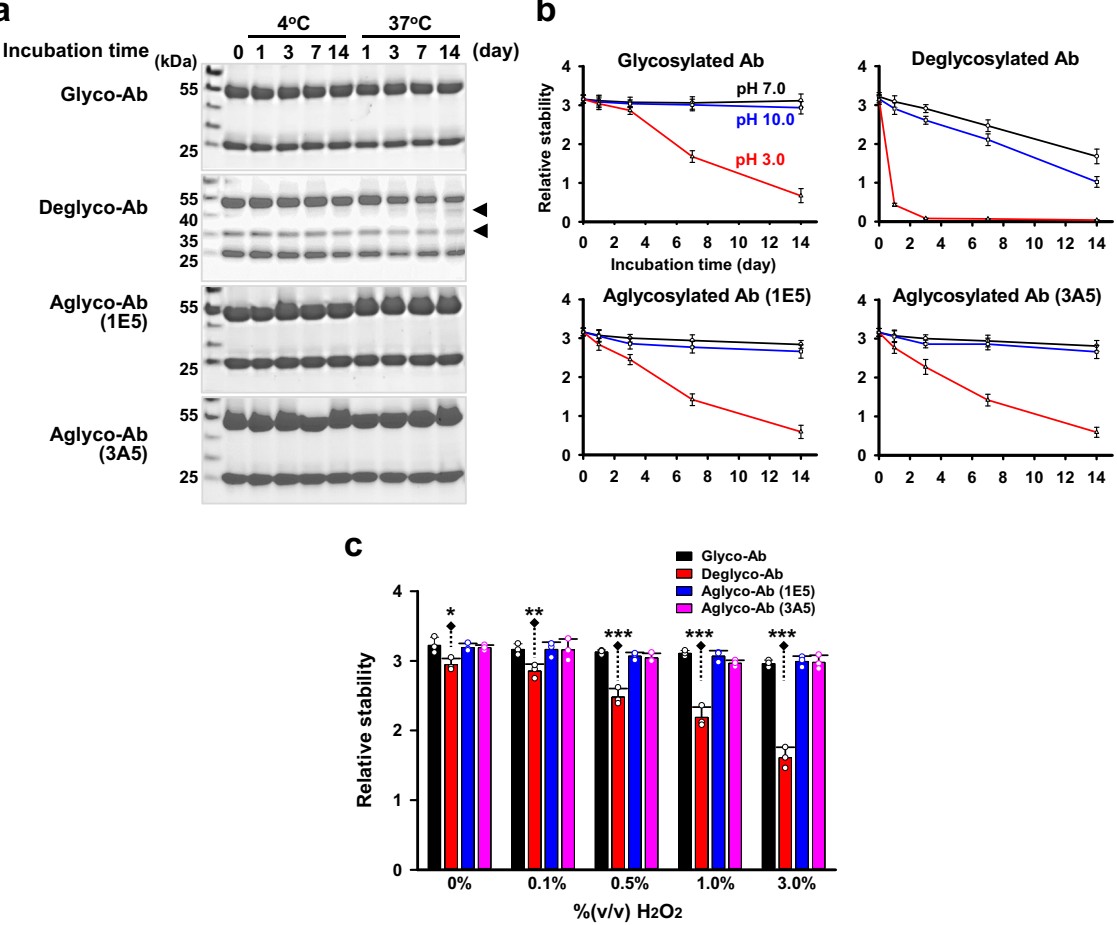

**Fig. 6 No compromised stability of aglycosylated antibodies. a** Preserved protein stability of aglycosylated antibodies during long-term incubation. Whereas the deglycosylation procedure by PNGase-F produced degradation products, aglycosylated antibodies retained integrity similar to that of glycosylated antibodies. **b** High resistance of aglycosylated antibodies to altered pH values. In contrast to a deglycosylated antibody, aglycosylated antibodies showed a similar pH-resistance pattern to that of a glycosylated antibody. Values are means ± standard deviations for triplicate measurements. **c** Retained integrity under oxidative conditions. Aglycosylated antibodies showed a negligible stability loss following incubations in PBS buffer containing 3.0% $H_2O_2$ for 1 h. *<0.05; **<0.01; ***<0.001, compared to glycosylated antibodies and aglycosylated antibodies; $n = 3$.

manner. In addition, there is no concern that the quality of antibodies deteriorates during the deglycosylation procedure.

It has become evident that aberrant glycosylation plays a multifaceted role in cancer development and progression, including invasion, proliferation, immune response, and angiogenesis[1,41]. Accordingly, the "sweet side" of tumor-specific glycoproteins has been extensively tested for the development of cancer drugs and in vitro diagnostics[41]. In particular, many FDA-approved cancer biomarkers are associated with changes in glycosylation structure through either incomplete or heavy glycosylation[2,3], as exemplified by CA125, human chorionic gonadotropin-β, CA 15-3, and CA27-29. AFP-L3 is a cancer-specific biomarker that is well characterized and clinically validated[42], and shows good discriminatory performance in the HCC/non-HCC blood test[43] and cancer staging[44]. As such, a robust automated analyzer for μ-Tas AFP-L3 tests has been developed and has secured FDA approval for staging of HCC[35]. Despite robustness and analytical validity, this platform has several limitations in universal and routine uses for glyco-biomarker analysis, as mentioned above. In contrast, CLIA or ELISA platforms that use an aglycosylated antibody and a glycoform-specific probe, such as a lectin, would provide more robust platform to measure tumor-specific glycoforms. Because aberrant glycosylation is widely associated with various pathological conditions, ALIQUAT can be expanded to various in vitro diagnostics. Collectively, our findings

suggest that our engineered mice will harness the development of precision in vitro diagnostics. In addition, basic researchers who struggle to relate the "sweet side" of glycoproteins to various disease states will also benefit.

Targeted gene correction can largely be achieved by either HDR-mediated sequence modifications using double-stranded DNA (dsDNA)-cutting programmable nucleases and donor DNA[45] or base-editing system consisting of engineered Cas proteins fused with base modifiers[19]. HDR-mediated gene correction shows less restriction than base editing in target selection because targets with the highest dsDNA breakage efficiency can be selected among possible candidates, and donor DNA can be designed considering a selected target. A major obstacle using this strategy is a relatively low HDR efficiency and this issue becomes more severe when using a limited number of zygotes. Fortunately, we have several approaches to improve HDR efficiency to some degree by, for instance, using an NHEJ pathway inhibitor[22] or an HDR-pathway activator[46]. The use of single-stranded oligonucleotides as donor DNA has been reported to improve HDR efficiency, compared to the use of dsDNA with relatively long homology arms[45]. Meanwhile, base-editing systems show relatively high modification efficiency[20]. A modification rate up to 100% has been reported by using both cytidine and adenine base-editing systems[47–50], which is significantly higher than HDR

efficiency (often <10%). In contrast, there are restrictions in target sequence because targeted modifications should be feasible within the base editing windows[19,51]. This restriction can be partly overcome by adopting xCas9 with a PAM flexibility[20]. In this report, we adopted both strategies and successfully edited IgG1 via HDR and *IgG2b*, *IgG2c*, and *IgG3* using an adenine base-editing system. To the best of our knowledge, this study is the first report in which stacked gene corrections were achieved by using both HDR and base-editing system. Recently, the concern of genome- and transcriptome-wide off-target activity was raised for cytidine base editors. Because adenine base editors turned out to be more specific than cytidine base editors[49,52], our engineered mice are thought to carry reduced pathogenic or harmful mutations that were newly occurring during genome engineering. Nonetheless, there still remained a possibility that unintended mutations could occur. This issue was addressed by several rounds of back-crossing with wild-type mice. Currently, we are conducting back-crossing of the engineered mice with BALB/c strain mice, the resultantly established mice would provide a more favorable host for the production of aglycosylated antibodies with a broader repertoire of antibodies[53].

## Methods

**Animals**. Procedures for the use and care of animals were reviewed and approved by the Institutional Animal Care and Use Committee, KRIBB. Zygotes were obtained from C57BL/6J male (8 weeks old) and female (4 weeks old) mice and ICR females (5 weeks old) were used for recipients. Engineered C57BL/6J male mice (8 weeks old) were back-crossed with BALB/c female mice (4 weeks old). All animals were bred in a specific pathogen-free facility with a constant temperature of 24 °C, humidity of 40%, and light cycles of 12 h[48].

**ABE mRNA transcription**. The pCMV-ABE7.10 and xCas9(3.7)-ABE(7.10) plasmid vectors were gifts from David Liu (Addgene, #102919 and Addgene, #108382, respectively). The plasmids were digested with AgeI (NEB) for 2 h at 37 °C and the linearized vectors were purified using a PCR purification kit (Qiagen). One microgram of the purified was used as a template for mRNA synthesis using an mMESSAGE mMACHINE T7 Ultra kit (Thermo Fisher Scientific). mRNA was isolated using the MEGAclear kit (Thermo Fisher Scientific) and aliquoted in cryotube vials prior to storage in liquid nitrogen.

**Preparation of donor DNA**. To achieve donor DNA preparation, genomic DNA was prepared from the tails of C57BL/6J mice. The Ighg1 locus with a 1.5-kb length was amplified using Pfu (Solgent) and cloned into a blunt vector using a T-Blunt™ PCR Cloning kit (Solgent). The vector construct was subjected to mutagenesis so that it contained the desired mutations, including glycosylation site, reporter sequence, and PAM mutations. The vector construct was PCR-amplified using the primers with desired mutations and the amplified products were treated with DpnI (NEB) at 37 °C for 1 h. The treated PCR products were used for transformation using DH5a BioFACT™ Competent cells (Biofact). The mutations were confirmed by Sanger sequencing analysis.

**Microinjection and electroporation**. hCG hormone (5 IU) was injected into the peritoneal cavity of C57BL/6J female mice (5 weeks old) at an interval of 48 h after PMSG hormone (5 IU, Merck). Female mice were mated with 9-week-old C57BL/6J male mice and one-cell zygotes were collected from the ampulla of the oviduct of female mice. Cumulus cells were removed by incubating in M2 media containing 3 mg/ml hyaluronidase (Merck). For HDR, the mixture comprising 100 ng/μl SpCas9 (Thermo Fisher Scientific), 100 ng/μl single-guide RNA (sgRNA) (Toolgen), 100 ng/μl dsDNA donor, and 1 mM Scr7 (Xcessbio) was micro-injected into the cytoplasm of zygotes using a Femtojet microinjector (Eppendorf) with an LEICA DMIRB manipulator (Leica Microsystems)[48]. For base editing, electroporation mixtures were prepared with 500 ng/μl sgRNA and 400 ng/μl ABE mRNA dissolved in opti-MEM (Gibco). Zygotes were suspended in the electroporation mixtures and subjected to electroporation in a NEPA 21 electroporator (NEPA GENE) according to the recommended protocol[54]. After microinjection or electroporation, zygotes were incubated in KSOM + AA medium (Millipore) at 37 °C in an incubator supplemented with 5% CO$_2$ and cultured until the two-cell stage. The viable cells were transplanted into the oviducts of pseudo-pregnant foster mice.

**Genotyping**. The toes were dissected from 1-week-old pups to isolate genomic DNA. The target loci were PCR-amplified using Pfu and specific primers (Supplementary Table 1). For HDR, the SacI restriction enzyme was applied to pre-screen the mutated pups. The final mutation was confirmed by Sanger sequencing of PCR products. For base editing, pre-screening of genotypes was conducted by

ARMS using H-Taq (Biofact). The primers used are listed in Supplementary Table 2. The samples with only a guanine band at the target site were subjected to Sanger sequencing analysis.

**Hybridoma cells**. Antigenic solutions were prepared by dissolving 50 μg of hAFP (Mybiosource) in 100 μl of TiterMax Gold adjuvant (Merck), and the emulsified solutions were injected into the footpad of engineered mice (6 weeks old) four times at an interval of 1 week each. The popliteal lymph nodes were dissected after the final boosting. B cells were collected and then fused with myeloma FO cells according to a previously described method[55]. A single fused cell was placed in a culture dish and cultured in Dulbecco's modified Eagle's medium (DMEM) culture medium supplemented with 20% fetal bovine serum, 1× HAT (100 μM hypoxanthine, 0.4 μM aminopterin, and 16 μM thymidine) (Merck), and 1× antibiotic–antimycotic solution (100 U/ml penicillin, 100 μg/ml streptomycin sulfate, and 0.25 μg/ml amphotericin B) (Welgene) for 2 weeks. Positive clones that produced antibodies against hAFP were screened using indirect ELISA tests. hAFP solutions were prepared by dissolving hAFP in PBS at a concentration of 1 μg/ml. Antigen solutions (100 μl) were used to coat the surface of 96-well plate (Thermo Fisher Scientific). After blocking, the culture-conditioned media were diluted by 100-fold and incubated in the coated well plates. After three rounds of washing with TBS-Tween-20 (0.02%), anti-mouse secondary antibody conjugated with horseradish peroxidase (HRP) (Cell Signaling Technology, diluted at 1:2000) were applied prior to treatment with TMB-ultra solution (Thermo Fisher Scientific). The chemical reaction was halted by adding 100 μl of 2 N H$_2$SO$_4$. The observance at 450 nm was measured using a VERSA max microplate reader (Molecular Devices). The positive clones were cultured in serum-free media (Gibco) to produce aglycosylated monoclonal antibodies.

**Production of monoclonal antibodies**. Hybridoma cells were cultured in serum-free media (Gibco) with agitation on a shaker at 100 RPM in a chamber supplemented with 5% CO$_2$ for 72 h. The conditioned media were collected, and cell debris were removed by centrifugation at 13,000 × g for 30 min. The remaining debris were finally removed by filtration through a 0.22-μm syringe filter (Millipore). The filtrates were purified on a HiTrap™ Protein G HP column (GE Healthcare Life Sciences) using an FPLC system (AKTA purifier, GE Healthcare Life Sciences). The column was equilibrated with PBS buffer at a flow rate of 1 min/ml, and bound proteins were eluted with 50 mM glycine-HCl (pH 2.5) at a flow rate of 3 ml/min. The eluted fractions were neutralized with 1 M Tris-HCl (pH 8.0) and then mixed with a protein-stabilizing cocktail solution (Thermo Fisher Scientific) prior to storage at −20 °C. Isotyping of the purified antibodies was performed using a mouse immunoglobulin isotyping kit (Antagen Pharmaceuticals). The protein concentration was adjusted to 1.0 μg/ml with PBS, and 50 μl of protein solutions were dropped into the sample-loading slots. The isotype was determined by the position of the red line formed.

**Mass analysis**. Ten micrograms of purified antibodies were resolved on 10% sodium dodecyl sulfate-polyacrylamide gel electrophoresis (SDS-PAGE) gels. The gels were stained with Bio-safe Coomassie solution (Bio-Rad) for 1 h and washed three times with distilled water for 30 min. The bands corresponding to large subunits were sliced into pieces. The sliced gels were lyophilized in a vacuum lyophilizer and treated with porcine trypsin (Promega) in 50 mM NaHCO$_3$ (pH 6.0) for in-gel digestion for 24 h. Mass analysis was performed at the Korea Basic Science Institute. Briefly, the digested peptides were retrieved from the gels by extractions with distilled water (twice) and acetonitrile. The trypsin-digested peptides were dissolved in a mobile phase solution and mass-analyzed using a liquid chromatography with tandem mass spectrometry (MS/MS) system consisting of a Nano Acquity UPLC system (Waters, USA) and an LTQ (linear trap quadrupole) Orbitrap Elite mass spectrometer (Thermo Scientific, USA) equipped with a nano-electrospray source. Five microliters of peptide solutions were loaded onto a C$_{18}$ trap column of i.d. 180 μm, length 20 mm, and particle size 5 μm (Waters, USA). After desalting and concentrating on the trap column, the trapped peptides were then separated on a homemade microcapillary C$_{18}$ column of i. d. 100 μm and length 200 mm (Aqua; particle size 3 μm, 125 Å with a gradient elution of mobile phases of 100% water and 100% acetonitrile, each containing 0.1% formic acid). A voltage of 2.2 kV was applied to produce an electrospray, and MS data were acquired using the following parameters: full scans were acquired in the Orbitrap at a resolution of 120,000 for each sample; six data-dependent collision-induced dissociation (CID) MS/MS scans were acquired per full scan; CID scans were acquired in a LTQ with 10 ms activation times performed for each sample; 35% normalized collision energy was used in CID; and a 2 Da isolation window for MS/MS fragmentation was applied. Previously, fragmented ions were excluded for 180 s.

***FUT8* knockout cells**. *FUT8*-targeting guide RNA (gRNA) constructs (Supplementary Table 3) were generated by cloning hybridized oligonucleotide pairs into the pSpCas9(BB)-2A-Puro (PX459) vector, which was a gift from Feng Zhang (Addgene plasmid #48139). Ten microgram of the vector was digested with 50 U of BbsI for 1 h, and digested vectors were gel-extracted using a gel-extraction kit (Solgent) and the paired gRNA inserts (10 pmol) were cloned into the linearized vector. Wild-type cell lines were obtained from ATCC. Knockout cell lines were generated in-house from the cell lines obtained from ATCC. All cell lines are

frequently tested for mycoplasma contamination. Cell lines used in this study were verified to be mycoplasma-free before undertaking any experiments with them. The target sequence was confirmed by Sanger sequencing. The vector constructs (10 μg) were used to transfect HEK293-T cells ($8 \times 10^5$ cells) (ATCC CRL-3216) using an electroporator (Neon, Invitrogen) with the following parameters: voltage 1300 V, pulse width 10 ms, and pulse number 3. A single cell was placed in each well to form a clone. Genotyping of each clone was performed by Sanger sequencing to screen the $FUT8^{-/-}$ clones.

**μ-Tas Wako AFP-L3 analysis**. AFP, AFP-L3, was measured in the remnant serum specimens using microchip capillary electrophoresis and a liquid-phase binding assay on an automatic analyzer (mTAS Wako i30, Wako Pure Chemical Industries, Osaka, Japan). The measurement range was 0.3–2000 ng/ml for AFP, and the AFP-L3 levels were calculated in sera whose AFP levels exceeded 0.3 ng/ml. If the AFP level of a sample was >2000 ng/ml, the original sample was manually diluted based on the previous results according to the manufacturer's instructions. All testing was conducted at the Pusan National University Yangsan Hospital by laboratory technicians, and none of the technicians were informed of the subject's status before testing.

**Immunofluorescence**. Wild-type engineered HEK293-T cells ($2 \times 10^4$) were allowed to grow for 1 day on 18 mm × 18 mm cover glass in DMEM media. Cells were fixed with BD cytofix/cytoperm solution (BD Bioscience) for 12 h. Cells were then incubated in the presence of 9 ng/μl PhoSL-Alexa Fluor 488 for 2 h at room temperature. After washing three times with PBS, cover glass was sealed in Vectashield mounting medium (Vector Laboratories) containing 1.5 μg/ml DAPI (4′,6-diamidino-2-phenylindole). Fluorescence was monitored on a Zeiss LSM510 Meta microscope (Carl Zeiss MicroImaging).

**Sandwich lectin-ELISA**. ELISA well plates were coated with 0.5 μg of either anti-hAFP mouse monoclonal antibody (#ab54745, Abcam) or our homemade anti-hAFP aglycosylated antibodies dissolved in 0.1 M sodium bicarbonate buffer (pH 7.4) at 4 °C overnight. The plates were washed twice with TBS-0.1% Tween-20 (TBS-T) and again with protein-free blocking buffer (Thermo Fisher Scientific) and 0.5% polyvinyl alcohol plus 0.1% Tween-20[56] for 1 h at room temperature. Either clinical samples or culture media (100 μl) were pipetted into each well for binding. Each well was washed with RIPA buffer (25 mM Tris-HCl 7.6, 150 mM NaCl, 1% NP-40, 1% sodium deoxycholate, 0.1% SDS) at least five times and again with TBS-T. Then, 100 μl of either an anti-hAFP rabbit polyclonal antibody (#ab8201, Abcam) diluted at 1:2000 or biotin-labeled PhoSL (#B-1395, Vector Laboratories) diluted at 1:1000 was added for 1 h at room temperature. After extensive washing, 1:2000 dilutions of an anti-rabbit secondary antibody (Cell Signaling) or streptavidin, each of which was conjugated with HRP, were added for 1 h at room temperature. After brief washing with TBS-T, 100 μl of a TMB substrate solution (Thermo Fisher Scientific) was added into each well, and the reactions were halted with 2 N sulfuric acid. The absorbance at 450 nm was measured.

**Human samples**. This study was approved by the Institutional Review Board at Pusan National University Yangsan Hospital (04-2017-023), and all procedures are carried out in accordance with the review board's guidelines and regulations. Volunteers and patients gave their informed written and oral consent for sample collections.

**Stability tests**. Deglycosylated antibody was prepared by incubating 10 μg of a commercially available anti-hAFP antibody (#MIA1305, Thermo Fisher Scientific) with 100 U of PNGase-F (NEB) in Rapid PNGase-F (non-reducing format) buffer for 15 min at 50 °C. Time- and temperature-dependent protein stability was measured by incubating glycosylated, deglycosylated, or aglycosylated antibodies in PBS buffer at either 4 or 37 °C for up to 14 days. The incubated antibodies were run on 4–20% Mini-Protein TGX$^{TM}$ gels (Bio-Rad) and visualized by Coomassie blue staining. The protein stability was also monitored by incubating antibodies in 0.1 M phosphate buffer adjusted to pH 3.0, 7.0, or 10.0 for 0–14 days. Alternatively, each antibody was incubated in PBS buffer containing 0–3% $H_2O_2$ at 37 °C for 5 h. The integrity of the antibodies was measured by the ELISA described above.

**Statistics and reproducibility**. Sample size was estimated based on a previous analysis by Lu et al.[57], which resulted in a total of 41 samples with a power of 80%, an alpha level of 0.05, different standardized difference limits of 3.0, and different standardized agreement limits of 0.2. Statistical significance tests were performed using Sigma Plot software (ver. 14.0) through a two-tailed Student's t test. P values <0.05 were considered significant. The ICC was measured to identify inter- and intra-rater repeatability. ICC estimates and their 95% CIs were calculated using MedCalc statistical software (MedCalc) based on a same-rater, absolute-agreement model. The reliability of a method was considered excellent if the lower estimate of the 95% CIs was ≥0.80. Bland–Altman agreement plots[37] were used to assess the agreement between two measurements. A Life Sciences Reporting Summary for this article is available.

**Reporting summary**. Further information on research design is available in the Nature Research Reporting Summary linked to this article.

## Data availability
The genome sequencing data were deposited at the NCBI Sequence Read Archive (http://www.ncbi.nlm.nih.gov/sra) under accession number PRJNA642339. The source data generated and/or analyzed in the current study are included in this article as Supplementary Data 2, which includes those for Figs. 1a, 3b, 4d–f, 5, and 6b, c. All other data that support the findings of the present study are available from the corresponding author upon request.

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

## Acknowledgements

This work was supported by the "Bio & Medical Technology Development Program" through the National Research Foundation and "KRIBB Research Initiative Program," funded by the Ministry of Science and ICT (NRF-2016M3A9B6903343), and "R&D Convergence Program" of the National Research Council of Science & Technology (CAP-15-03-KRIBB).

## Author contributions

Y.-S.K. and J.-H.K. conceived the study and designed the experiments. N.-E.L. and S.H.K. performed the overall experiments. M.-I.K. and G.-S.S. performed an immunoassay. S.M.L. performed the μ-Tas analysis. D.-Y.Y. performed mouse engineering. E.-J.W. performed protein structural analysis. Y.-S.K. wrote the manuscript.

## Competing interests

The authors declare no competing interests.
