## [Peer Review File · Communications Biology]

Reviewers' comments:

Reviewer #1 (Remarks to the Author):

In this manuscript, Lee et al describe the generation of aglycosylated antibody-producing mice. The author reasoned that the aglycosylated antibodies, lack glycans which prevent unwanted interactions of the antibody with the lectins, could be very useful reagents in lectin-based immunoassay diagnostic platform detecting disease specific glycan biomarkers. The ALIQUAT method that they have come up with, using the antibody produced from those mice, is of clear interest and the assay shows some advancement, as shown in their feasibility test.

Major concern:

1. It appears to me the authors have ignored IgG glycosylation that are present on the other parts of the antibodies, for example Fab glycans. Historically, ASN-297 Nglycosylation of the CH2 domain of the IgG has been extensively studied and paid little attention to the glycosylation present on the other parts of the antibodies, I see the same trend in this manuscript except briefly mentioning about Fab glycans in the result section ref 29. Interestingly, recent emerging work have increasingly shown interest in Fab glycans as referenced in 29, so we will witness more and more studies along this

line. The authors should consider briefly updating the literature in their introduction about the Fab glycosylation and pay extra attention on the potential impacts of those glycans on their assay and their result interpretation.

2. The authors should consider using FC aglycosylated antibody instead of using aglycosylated antibody in the manuscript; my understanding for the correct use of the term aglycosylated antibody is for those antibodies that are being expressed and purified using bacterial system or remove the glycans exogenously.

3. Line79 -the author should consider rewording/deleting- fundamental manner.

4. Line 102 the author should consider deleting or rewording-Once-for all/ once for ever strategy for the same reason as explained above- briefly IgGs may contain glycosylation on the other part of the molecule and lectin probe may bind to those glycans generating nonspecific background. It's important to point out that Lectins are known for nonspecific interactions.

5. The authors confirm that using ConA lectin as shown in Figure 3E and Supplementary Figure 8, their antibody is aglycosylated, these experiments required to be performed with appropriate controls. PNGase F treatment should remove ConA staining seen on IgG. Also, glycoprotein like fetuin could be used as a positive control.

6. Labelling of all the markers in blots is required for all the figures.

7. Figure 4B. In this immunofluorescence study, please also include PhoSL staining preincubated with fucose which, in principle, should inhibit staining.

8. Figure 4 C. I am assuming that the size of the bands corresponds to the expected size. Would you see drop in size of AFP-L3 to AFP-L1 if you treat AFP-L3 (expressed in HEK293 T cells or serum) with α 1,6-Fucosidase in your lectin affinity electrophoresis?

9. Figure 4e and f. Did you consider using fucose to inhibit binding?

Minor comments:

1. Line 264 to those should be to those.

2. Figure 2, Please define the abbreviation for eg, NC, PC, NP, and WT in the figure legend in all figures.

Reviewer #2 (Remarks to the Author):

1. Brief summary of the manuscript

Lee et al. generated aglycosylated antibody-producing mice by genome engineering and established an aglycosylated antibody-lectin coupled immunoassay for the quantification of tumor markers (ALIQUAT) to circumvent false-positive detection via non-specific carbohydrate chains on the Fc region of anti-biomarker (anti-human AFP in this study) IgG antibodies. First, homozygous D-S-T mutations were introduced to the N-S-T position, which is responsible for the N-glycosylation in IgG2b, IgG2c, and IgG3 locus using an adenine base-editing system to generate the aglycosylated antibody-producing mice. Similarly, the N-S-T position in the IgG1 locus was substituted with E-L-T mutation using HDR system. After that, the authors tested the engineered mouse for unintended off-target activity by conducting several rounds of back-crossing to clear any potentially harmful mutations. Finally, they confirmed that the engineered mice not only bears sequences that do not produce N glycan i.e. either D-S-T or E-L-T in the genes of all the IgG subclasses but also produced completely aglycosylated IgG antibodies.

To test the feasibility of the aglycosylated antibodies for lectin-ELISA the authors compared the standard curves of both total AFP and AFP-L3 binding using the aglycosylated antibodies generated in this study with those using commercial glycosylated antibodies. As a result, the lectin-ELISA using PhoSL (a lectin known to bind specifically to fucosylated N-glycan i.e. AFP-L3) and aglycosylated antibody produced a good linearity standard curve without interference in the range of 0-100ng/ml. Therefore, the lectin-ELISA using the aglycosylated anti-AFP antibody was proven to be feasible for detecting specific glycoforms. Compared to the result μ -TAS AFP-L3 analysis, the ALIQUAT method provides a reliable analytical platform for AFP/AFP-L3 tests and could be replaced with μ -TAS AFP-L3 analysis.

Additionally, the aglycosylated antibodies exhibited similar stability profiles to the conventional N-glycosylated antibody under different conditions of temperature, pH, or oxidation, but showed superior to the deglycosylated antibody of which the N-glycans were removed enzymatically.

2. Overall impression of the work

I would recommend this manuscript for acceptance to the Communications Biology if the authors can clearly state that the method in this study using the engineered mice is advantageous over other strategies; for example, i) directly substituting amino acid residue Asn with others, followed by Ser-Thr that is critical for N-glycosylation in the existing anti-biomarker (AFP in this case) IgG antibodies to generate aglycosylated IgG antibodies or ii) using single-chain Fv (scFv) format, which does not include any glycan chain.

In addition, it would be great if the authors can list other applicable examples that these engineered mice could be used.

3. Specific comments, with recommendations for addressing each comment

Comment 1. To validate that the lectins used in this study are AFP-specific the authors should show that the lectins do not bind to the aglycosylated IgG, as opposed to glycosylated counterpart. This is important because the purpose of using the aglycosylated Abs in this study was to circumvent non-specific binding of lectin to carbohydrate chains on IgG, especially when antibodies can be applicable to HCC diagnosis.

Comment 2. Since the glycosylated Ab is a commercial one, it is highly possible that the Fab regions of aglycosylated Ab and glycosylated Ab are different in Figures 4e and f: it was best if the Fab region of the glyco- and aglyco- forms were the same. If this were the case, do you expect that the glycan chain will affect total AFP or AFP-L3 binding? In the same sense, do you think the

E-L-T mutation would affect AFP or AFP-L3 binding?

Comment 3. In the IgG production test, Line 198-199 states that the engineered mice displayed an identical profile to that of a wild-type C57BL/6 mouse; however, Figure 3a does not look the same, especially for IgG2c. Can you corroborate on this in the sense that production of every subclass of IgG from engineered mice was concluded in this study.

Comment 4. The thermostability of IgG antibodies were tested using SDS-PAGE in Figure 6a, where the protein solubility and/or activity cannot be distinguished. It would be best if they were measured by ELISA just as in pH- or oxidative- stress conditions (Figure 6b and c).

Comment 5. A few typing errors were found in this paper.

Line 128: double strand → double stranded

Line 134: IgG2g → IgG2b

Line 214: HAS → HSA

Line 264: tothose → to those

Line 648-649: The format of reference 18 should be corrected.

Comment 7. Misuse of aglycosylation is in the discussion section.

Using of PNGase-F is deglycosylation method. But, in discussion section, PNGase-F was used as an example of aglycosylation.

Reviewer #3 (Remarks to the Author):

Summary of the paper titled "Generation of aglycosylated antibody-producing mice for aglycosylated antibody-lectin coupled immunoassay for the quantification of tumor markers (ALIQAT)".

In this paper, the authors have developed a new tool called ALIQAT which stands for "aglycosylated antibody-lectin coupled immunoassay for the quantification of tumor markers". The need for this new technique arises from high background levels and error rates received with glycosylated antibodies in ELISA assays, due to glycoform specific lectins binding to complex N-glycans on antibodies rather than the analytes/glycoproteins in tumor samples.

To eliminate the unwanted Lectin binding to antibodies, the authors mutated the key N-glycan sequons on all four IgG genes to an amino acid that cannot be glycosylated, which they call the Phase 1 of their study. After successfully generating the mouse with four different mutations and no background mutation in the rest of the genome, they proceed to establishing the hybridoma cell line for the production of antibodies, Phase 2. Last, they test the method ALIQAT in comparison to μ -TAS AFP-L3 (existing FDA approved test for AFP testing only) and show significant results with a 95%CI. Finally, they also look into the stability of the aglycosylated antibodies under various conditions, temperature, pH, and oxidative and reveal that lack of glycosylation does not affect the stability of their antibodies. Overall, the work is intriguing and novel. The mouse lines generated here should have wide applications in antibody generation for diagnostics.

Comments:

The strategy to generate the genetically engineered mouse line is logical, enabling a single source of antibody] production reducing the heterogeneity i.e. compared to mutating harvested cells. They artfully apply Adenine Base editing sequentially with HDR to achieve all four mutations in IgG genes, successfully. The authors validate their results at each stage with carefully designed PCR assays and Sanger sequencing. Finally, they cross out all the unwanted off target mutations, which is ideal. The authors produce AFP-L1 and AFP-L3 samples for the feasibility test of aglycosylated antibodies using Fut8^{-/-} mutant HEK cells and use these to test their aglycosylated antibodies in comparison to commercially available antibodies. This shows greatly reduced background levels

with the ALIQUAT method which is convincing. There are a few minor issues that should be addressed to render the paper suitable to the readership of "Communications Biology":

- Is the ELISA data in Fig. 1a actual new data taken in this study?
- The gel in Figure 2c has a somewhat lighter region in which the major bands are seen. Could the authors please explain this?
- The authors should clarify the very faint band in Figure 3e; E1 lane that shows binding with the lectin Concanavalin-A.
- The differences in binding curves in Figures 4e and 4f are very convincing, but it would be great to have the same binding data with deglycosylated commercial antibody. If this has been done in a different paper, it would be good to highlight this here.
- Mass spec data should be provided for the absence of the AFP-L3 glycoform in FUT8^{-/-} cells. This is so far only mentioned in the manuscript.
- Are the mouse line and the mAbs generated here free of charge, subject to an MTA, or being commercialized?

Issues with the text:

- The Summary needs another sentence on cross-reactivity of glycosylated Abs with lectins.
 - The sentence "In fact, glycoproteins account for a large portion of FDA-approved cancer biomarkers, and furthermore, an altered glycan structure or a specific glycoform is considered as an analyte for in vitro cancer diagnostics." Should be two sentences.
 - Line 44, "One of the best characterized biomarkers is..." should be "best characterized cancer biomarkers".
 - Line 53 The following sentence is rather complicated and should be re-phrased: "More recently, an immunoassay to measure the Wisteria Floribunda agglutinin-positive Mac-2 binding protein glycan isomer (M2BPGi) was developed to monitor patients with liver fibrosis and cirrhosis and to predict the development of HCC in hepatitis B patients treated with nucleot(s)ide analogues."
 - Line 87 "assay results substantially identical to" – leave out "substantially"
 - Line 100 "glycan structure is not usually easy to predict" should be "is often difficult to"
 - Line 102 leave out "radically"
 - Line 149 "The finally generated pups carried one of the mutant amino acid motifs, D-S-T, D-G-T, G-S-T, and G-G-T." this should specify which isotype the sequences refer to.
 - Line 203 "we attempted to produce" should be "we produced"
 - Figure 4e Label for y axis is missing.
 - Fig. 4b can the authors please include the brightfield image here?
 - Line 288 "Forty-one randomly selected 41 clinical samples" leave out "41"
 - Bland-Altman analysis should be briefly explained. This is especially important as the cut-off values of 2% and 10% mentioned in the next are not directly depicted in Fig. 5d and 5e.
 - In Fig 6a, molecular weight markers should be shown.
 - Line 214 "reactivity for HSA" should be "reactivity for HAS"
 - Line 264 Insert space between to and those "tothose"
 - Line 368 Sentence "Researchers or commercial vendors have only to use engineered mice to directly produce aglycosylated antibodies without in vitro enzymatic reactions and purification." should be corrected as "Researchers or commercial vendors have to only use (OR "only have to use")engineered mice to directly produce aglycosylated antibodies without in vitro enzymatic reactions and purification."
 - Line 411 Should say "Cytosine base editor" instead of "Cytidine base editor".
- Experimentals:
- Line 428 "humility" should be "humidity"
 - Addgene vectors should be appropriately acknowledged, including the depositing author, as specified on the addgene website.
 - Primer sequences for site-directed mutagenesis and standard cloning should be included.
 - Line 484 TMB is quenched with "0.2 M H₂SO₄" while on line 561 with "2 N sulfuric acid".

- Line 543 Should be DMEM media instead of DEME?
- Line 549 Elisa well plates should be specified, cat number?
- Line 56 "TMP" should be "TMB".

Comments on the Supplementary Information

Supp. Figure 6. Ladder should be labelled on the gel.

Supp. Figure 8. It would be helpful to see a higher exposure of the Concanavalin-A blot.

Reviewers' comments:

Reviewer #1 (Remarks to the Author):

We are grateful for the fruitful comments by reviewer 1. We revised our manuscript in compliance with the reviewer's suggestion. We estimate that the revised manuscript was improved for attracting broader readership and providing additional information to readers. Thank you for your time and comments.

In this manuscript, Lee et al describe the generation of aglycosylated antibody-producing mice. The author reasoned that the aglycosylated antibodies, lack glycans which prevent unwanted interactions of the antibody with the lectins, could be very useful reagents in lectin-based immunoassay diagnostic platform detecting disease specific glycan biomarkers. The ALIQUAT method that they have come up with, using the antibody produced from those mice, is of clear interest and the assay shows some advancement, as shown in their feasibility test.

Major concern:

1. It appears to me the authors have ignored IgG glycosylation that are present on the other parts of the antibodies, for example Fab glycans. Historically, ASN-297 N-glycosylation of the CH₂ domain of the IgG has been extensively studied and paid little attention to the glycosylation present on the other parts of the antibodies, I see the same trend in this manuscript except briefly mentioning about Fab glycans in the result section ref 29. Interestingly, recent emerging work have increasingly shown interest in Fab glycans as referenced in 29, so we will witness more and more studies along this line. The authors should consider briefly updating the literature in their introduction about the Fab glycosylation and pay extra attention on the potential impacts of those glycans on their assay and their result interpretation.

As the reviewer indicated, recent works tell us about the importance of Fab glycosylation. It is likely that a portion of N-X-S/T motifs in Fab are N-glycosylated when structurally compatible with reaction by glycosyltransferases. We were aware of the fact and that's why we investigated the glycosylation status of our aglycosylated antibodies. Nonetheless, we thought that we need to more stress whether our antibodies are completely free of glycosylation even in Fab. For this aim, we revised the Supplementary Figure 8 by adding a positive control and investigating glycosylation focused on Fab fragments. We also added the recent literature on Fab glycosylation in the result section.

2. The authors should consider using FC aglycosylated antibody instead of using aglycosylated antibody in the manuscript; my understanding for the correct use of the term aglycosylated antibody is for those antibodies that are being expressed and purified using bacterial system or remove the glycans exogenously.

As you suggested, we named our engineered antibodies as an Fc-aglycosylated antibody at the initially mentioned sentence. Then, aglycosylated antibody was instead used for simplicity. To differentiate our antibodies from the processed ones, antibodies whose glycans are removed exogenously were termed deglycosylated antibody in our manuscript.

3. Line79 -the author should consider rewording/deleting- fundamental manner.

We corrected the sentence as "To resolve these problems straightforward,".

4. Line 102 the author should consider deleting or rewording-Once-for all/ once for ever strategy for the same reason as explained above- briefly IgGs may contain glycosylation on the other part of the molecule and lectin

probe may bind to those glycans generating nonspecific background. It's important to point out that Lectins are known for nonspecific interactions.

The phrase was also reworded as "straightforward".

5. The authors confirm that using ConA lectin as shown in Figure 3E and Supplementary Figure 8, their antibody is aglycosylated, these experiments required to be performed with appropriate controls. PNGase F treatment should remove ConA staining seen on IgG. Also, glycoprotein like fetuin could be used as a positive control.

As mentioned in Question 1, we revised the Supplementary Figure 8 by adding more glycosylated antibodies as a positive control and BSA as a negative control to investigate glycosylation features. Those results confirmed that Con-A-based lectin blot provided a robust platform for detecting N-glycans and that our Fc-aglycosylated antibody does not carry any N-glycan.

6. Labelling of all the markers in blots is required for all the figures.

Labeling of molecular markers were included in Fig. 2a-2d, Fig. 6a, Supplementary Fig. 6a, and 8.

7. Figure 4B. In this immunofluorescence study, please also include PhoSL staining preincubated with fucose which, in principle, should inhibit staining.

Inhibition assay using L-fucose as a competing agent revealed that L-fucose did not show any inhibition profile toward PhoSL-core fucose binding. In fact, the first paper on the identification of PhoSL published in JBC (Kobayashi et al., 2012) reported that no monosaccharides including fucose did not inhibit PhoSL binding to core fucose.

8. Figure 4 C. I am assuming that the size of the bands corresponds to the expected size. Would you see drop in size of AFP-L3 to AFP-L1 if you treat AFP-L3 (expressed in HEK293 T cells or serum) with α 1,6-Fucosidase in your lectin affinity electrophoresis?

Unfortunately, we could not observe any changes in the mobility of AFP-L3 on lectin affinity electrophoresis gels even after treatments with α 1,6-fucosidase. There are two possibilities for this result. The fucose residue on AFP is not accessible by α 1,6-fucosidase, or the α 1,6-fucosidase we have in our lab is non-functional. We tested with two different batches of α 1,6-fucosidase, but we obtained the same results. Another approach would be perform in-gel digestion of AFP-L3 with α 1,6-fucosidase and trypsin, and the produced glycopeptides can be analyzed by mass spectrometry. We've seen a delay in delivery of α 1,6-fucosidase recently because of the corona pandemic issue, so we won't be able to incorporate the related study. Please understand this situation, and we will test the approach when it's available.

9. Figure 4e and f. Did you consider using fucose to inhibit binding?

As mentioned in Q7, fucose does not inhibit PhoSL binding to core fucose. Thus, we did not consider the experiment this time you suggested.

Minor comments:

1. Line 264 tothose should be to those.

The typo was corrected.

2. Figure 2, Please define the abbreviation for eg, NC, PC, NP, and WT in the figure legend in all figures.

All the abbreviations were defined in the figure legend.

Reviewer #2 (Remarks to the Author):

We are grateful for the fruitful comments by reviewer 2. We revised our manuscript in compliance with the reviewer's suggestion. We estimate that the revised manuscript was improved for attracting broader readership and providing more correct information to readers. Thank you for your time and comments.

1. Brief summary of the manuscript

Lee et al. generated aglycosylated antibody-producing mice by genome engineering and established an aglycosylated antibody-lectin coupled immunoassay for the quantification of tumor markers (ALIQAT) to circumvent false-positive detection via non-specific carbohydrate chains on the Fc region of anti-biomarker (anti-human AFP in this study) IgG antibodies. First, homozygous D-S-T mutations were introduced to the N-S-T position, which is responsible for the N-glycosylation in IgG2b, IgG2c, and IgG3 locus using an adenine base-editing system to generate the aglycosylated antibody-producing mice. Similarly, the N-S-T position in the IgG1 locus was substituted with E-L-T mutation using HDR system. After that, the authors tested the engineered mouse for unintended off-target activity by conducting several rounds of back-crossing to clear any potentially harmful mutations. Finally, they confirmed that the engineered mice not only bears sequences that do not produce N glycan i.e. either D-S-T or E-L-T in the genes of all the IgG subclasses but also produced completely aglycosylated IgG antibodies.

To test the feasibility of the aglycosylated antibodies for lectin-ELISA the authors compared the standard curves of both total AFP and AFP-L3 binding using the aglycosylated antibodies generated in this study with those using commercial glycosylated antibodies. As a result, the lectin-ELISA using PhoSL (a lectin known to bind specifically to fucosylated N-glycan i.e. AFP-L3) and aglycosylated antibody produced a good linearity standard curve without interference in the range of 0-100ng/ml. Therefore, the lectin-ELISA using the aglycosylated anti-AFP antibody was proven to be feasible for detecting specific glycoforms. Compared to the result μ -TAS AFP-L3 analysis, the ALIQAT method provides a reliable analytical platform for AFP/AFP-L3 tests and could be replaced with μ -TAS AFP-L3 analysis.

Additionally, the aglycosylated antibodies exhibited similar stability profiles to the conventional N-glycosylated antibody under different conditions of temperature, pH, or oxidation, but showed superior to the deglycosylated antibody of which the N-glycans were removed enzymatically.

2. Overall impression of the work

I would recommend this manuscript for acceptance to the Communications Biology if the authors can clearly state that the method in this study using the engineered mice is advantageous over other strategies; for example, i) directly substituting amino acid residue Asn with others, followed by Ser-Thr that is critical for N-glycosylation in the existing anti-biomarker (AFP in this case) IgG antibodies to generate aglycosylated IgG antibodies or ii) using single-chain Fv (scFv) format, which does not include any glycan chain. In addition, it would be great if the authors can list other applicable examples that these engineered mice could be used.

The reviewer mentioned two alternative methods to create aglycosylated IgG antibodies against a specific biomarker: a direct substitution of Asn with non-Asn residues and a use of single-chain Fv (scFv). The two engineering methods can be actually employed to create an aglycosylated antibody, but what differs with our strategy is this: Those two engineering works should be performed every time an individual antibody is produced. That is, the fusion of V_H and V_L should be done for individual antibodies. In addition, a direct substitution of Asn with others can be performed in hybridoma cells, but the substitution works should also be done for each antibody. That is, those two approaches requires engineering efforts on an individual antibody basis, and thus, are very time-consuming. However, researchers would have only to produce antibodies from our engineered mice without further engineering steps to obtain aglycosylated antibodies. That's why we described our approach as "once-for-all/once-forever strategy". These were complemented in Discussion.

One more application would be that the engineered mice can be used as a model animal during non-clinical trials. Biological therapeutics are predominantly tested in mice in terms of PK and PD. More often than not, biologics, often humanized ones, trigger immune responses in model mice and thus, are immediately cleared through evoked antibody productions, which makes it difficult to achieve precise PK assessments. Such an antibody-dependent

clearing process is mediated by IgG N-glycans. Because our engineered mice are expected to show mitigated clearing responses during PK study, they would be preferably used for PK studies of various biologics as model mice. Currently, we are investigating the relevant POC tests.

3. Specific comments, with recommendations for addressing each comment

Comment 1. To validate that the lectins used in this study are AFP-specific the authors should show that the lectins do not bind to the aglycosylated IgG, as opposed to glycosylated counterpart. This is important because the purpose of using the aglycosylated Abs in this study was to circumvent non-specific binding of lectin to carbohydrate chains on IgG, especially when antibodies can be applicable to HCC diagnosis.

The lectin blot analysis using PhoSL was implemented using commercial antibodies as a positive control and bovine serum albumin as a negative control, confirming that our aglycosylated antibody showed no interaction with PhoSL. The related results were presented in Supplementary Fig. 8.

Comment 2. Since the glycosylated Ab is a commercial one, it is highly possible that the Fab regions of aglycosylated Ab and glycosylated Ab are different in Figures 4e and f: it was best if the Fab region of the glyco- and aglyco- forms were the same. If this were the case, do you expect that the glycan chain will affect total AFP or AFP-L₃ binding? In the same sense, do you think the E-L-T mutation would affect AFP or AFP-L₃ binding?

It is generally regarded that the glycan chains in Fc region do not affect the antigen-antibody interactions and that the antigen-antibody interaction mostly depends on the structures of variable chains. As you mentioned, if our engineered antibodies had the same Fab sequence as that of the tested commercial one, a more direct comparison would be possible and that our ALIQUAT method for AFP/AFP-L₃ measurements would show higher analytical sensitivity. However, there's no sequence information on Fab regions of the commercial antibody. It is likely that the Fab sequences of our aglycosylated antibodies are less optimal, compared to the commercial one. Thus, we currently attempt to screen a monoclonal antibody with a similar or even higher affinity, compared to the commercial one. In fact, we tested nearly ~1,000 hybridoma clones, all of which carry the same E-L-T sequence in Fc region. However, each aglycosylated antibody clone shows different affinity toward AFP, indicating that the Fab sequence is likely to govern an affinity. If we screen a better aglycosylated antibody, we expect that our ALIQUAT method would produce better analytical performances.

Comment 3. In the IgG production test, Line 198-199 states that the engineered mice displayed an identical profile to that of a wild-type C57BL/6 mouse; however, Figure 3a does not look the same, especially for IgG_{2c}. Can you corroborate on this in the sense that production of every subclass of IgG from engineered mice was concluded in this study.

It is likely that there are individual-to-individual variations in the profile of IgG expressions. We performed IgG profiling in additional engineered mice and found that their profiles are close to those of wild-type mice. An IgG₃ band was less clearer than other IgG subclasses but the pattern was identically seen in both WT and engineered mice. We attach an additional profiling image below.

Comment 4. The thermostability of IgG antibodies were tested using SDS-PAGE in Figure 6a, where the protein solubility and/or activity cannot be distinguished. It would be best if they were measured by ELISA just as in pH- or oxidative- stress conditions (Figure 6b and c).

In Fig. 2b, the thermostability test was already performed through ELISA tests. The results of pH 7.0 correspond to the thermostability test.

Comment 5. A few typing errors were found in this paper.

Line 128: double strand → double stranded

Line 134: IgG2g → IgG2b

Line 214: HAS → HSA

Line 264: tothose → to those

Line 648-649: The format of reference 18 should be corrected.

All typos were corrected in the manuscript

Comment 7. Misuse of aglycosylation is in the discussion section.

Using of PNGase-F is deglycosylation method. But, in discussion section, PNGase-F was used as an example of aglycosylation.

"Aglycosylation of an antibody" was changed to "Glycan-free antibody" to prevent confusions in the meaning of aglycosylation.

Reviewer #3 (Remarks to the Author):

We are grateful for the fruitful comments of reviewer 3. We revised our manuscript in compliance with the reviewer's suggestion. We estimate that the revised manuscript was improved for attracting broader readership and providing information with integrity to readers. Thank you for your time and comments.

Summary of the paper titled "Generation of aglycosylated antibody-producing mice for aglycosylated antibody-lectin coupled immunoassay for the quantification of tumor markers (ALIUQUAT)".

In this paper, the authors have developed a new tool called ALIUQUAT which stands for "aglycosylated antibody-lectin coupled immunoassay for the quantification of tumor markers". The need for this new technique arises from high background levels and error rates received with glycosylated antibodies in ELISA assays, due to glycoform specific lectins binding to complex N-glycans on antibodies rather than the analytes/glycoproteins in tumor samples. To eliminate the unwanted Lectin binding to antibodies, the authors mutated the key N-glycan sequons on all four IgG genes to an amino acid that cannot be glycosylated, which they call the Phase 1 of their study. After successfully generating the mouse with four different mutations and no background mutation in the rest of the genome, they proceed to establishing the hybridoma cell line for the production of antibodies, Phase 2. Last, they test the method ALIUQUAT in comparison to μ -TAS AFP-L3 (existing FDA approved test for AFP testing only) and show significant results with a 95%CI. Finally, they also look into the stability of the aglycosylated antibodies under various conditions, temperature, pH, and oxidative and reveal that lack of glycosylation does not affect the stability of their antibodies. Overall, the work is intriguing and novel. The mouse lines generated here should have wide applications in antibody generation for diagnostics.

Comments:

The strategy to generate the genetically engineered mouse line is logical, enabling a single source of antibody production reducing the heterogeneity i.e. compared to mutating harvested cells. They artfully apply Adenine Base editing sequentially with HDR to achieve all four mutations in IgG genes, successfully. The authors validate their results at each stage with carefully designed PCR assays and Sanger sequencing. Finally, they cross out all the unwanted off target mutations, which is ideal. The authors produce AFP-L1 and AFP-L3 samples for the feasibility test of aglycosylated antibodies using Fut8^{-/-} mutant HEK cells and use these to test their aglycosylated antibodies in comparison to commercially available antibodies. This shows greatly reduced background levels with the ALIUQUAT method which is convincing. There are a few minor issues that should be addressed to render the paper suitable to the readership of "Communications Biology":

- Is the ELISA data in Fig. 1a actual new data taken in this study?

The data were actual results initially obtained while performing experiments of Fig. 4f.

- The gel in Figure 2c has a somewhat lighter region in which the major bands are seen. Could the authors please explain this?

The result in Figure 2c is the negative images of original pictures of gels. We add the original image for Fig. 2c below. In the original image, we can found that the darker regions in the original images (corresponding to lighter region in the negative image in Fig. 2c) arised from loading dyes.

- The authors should clarify the very faint band in Figure 3e; E1 lane that shows binding with the lectin Concanavalin-A.

The faint band in the e1 lane is likely to stem from a fraction of samples overflowed from the "media" lane. The e2 lane shows no such band even though more IgG proteins were loaded the lane.

- The differences in binding curves in Figures 4e and 4f are very convincing, but it would be great to have the same binding data with deglycosylated commercial antibody. If this has been done in a different paper, it would be good to highlight this here.

As is presented in Fig. 6b, deglycosylated commercial antibodies showed a significant impairment in the stability and thus binding property as a capture antibody even under neutral pH conditions. For this reason, data obtained using deglycosylated antibodies were severely scattered without focusing on an average value. Thus, we determined that an additional incorporation of data using deglycosylated antibodies was not necessary.

- Mass spec data should be provided for the absence of the AFP-L3 glycoform in FUT8-/- cells. This is so far only mentioned in the manuscript.

In Supplementary Figure 9, MS spectra and chromatograms are provided for several representative glycoforms with a peptide sequence of VNFTEIQK. The peaks in chromatogram for each fucosylated glycoform are marked with an arrow. The absence of fucosylated glycoform in KO cell lines can be confirmed with an absence of peaks in the same retention time in chromatograms of wild-type samples. Because of the limitations in space, representative glycopeptides were presented in Supplementary Figure 9. We manually confirmed that all the other fucosylated glycoforms were not identified in FUT8-/- cells.

- Are the mouse line and the mAbs generated here free of charge, subject to an MTA, or being commercialized?

We have no fixed plan for that for now. But we are searching for companies for licensing out of our engineered mice. Only before that, we may be able to provide monoclonal clones with minimal operation costs with an MTA preceded.

*** Issues with the text:**

- The Summary needs another sentence on cross-reactivity of glycosylated Abs with lectins.

- The sentence "In fact, glycoproteins account for a large portion of FDA-approved cancer biomarkers, and furthermore, an altered glycan structure or a specific glycoform is considered as an analyte for in vitro cancer diagnostics." Should be two sentences.

- Line 44, "One of the best characterized biomarkers is..." should be "best characterized cancer biomarkers".

- Line 53 The following sentence is rather complicated and should be re-phrased: "More recently, an immunoassay to measure the Wisteria Floribunda agglutinin-positive Mac-2 binding protein glycan isomer (M2BPGi) was developed to monitor patients with liver fibrosis and cirrhosis and to predict the development of HCC in hepatitis B patients treated with nucleot(s)ide analogues."

- Line 87 "assay results substantially identical to" – leave out "substantially"

- Line 100 "glycan structure is not usually easy to predict" should be "is often difficult to"

- Line 102 leave out "radically"

- Line 149 "The finally generated pups carried one of the mutant amino acid motifs, D-S-T, D-G-T, G-S-T, and G-G-T." this should specify which isotype the sequences refer to.

- Line 203 "we attempted to produce" should be "we produced"

- Figure 4e Label for y axis is missing.

- Fig. 4b can the authors please include the brightfield image here?

- Line 288 "Forty-one randomly selected 41 clinical samples" leave out "41"

- Bland-Altman analysis should be briefly explained. This is especially important as the cut-off values of 2% and 10% mentioned in the next are not directly depicted in Fig. 5d and 5e.

- In Fig 6a, molecular weight markers should be shown.

- Line 214 "reactivity for HSA" should be "reactivity for HAS"

- Line 264 Insert space between to and those "tothose"

- Line 368 Sentence "Researchers or commercial vendors have only to use engineered mice to directly produce aglycosylated antibodies without in vitro enzymatic reactions and purification." should be corrected as "Researchers or commercial vendors have to only use (OR "only have to use")engineered mice to directly produce aglycosylated antibodies without in vitro enzymatic reactions and purification."
- Line 411 Should say "Cytosine base editor" instead of "Cytidine base editor".

All questions, except for brightfield image in Fig. 4b, were addressed appropriately in the revised manuscript and the corrections were highlighted with red letters. Unfortunately, we could not find the original brightfield images saved in the previous works, and newly performing immunofluorescence requires an import of PhoSL. However, the corona pandemic issue made it difficult for us to obtain the validated commercialized version of PhoSL. We feel sorry for not responding to this claim.

- Experimentals:

- Line 428 "humility" should be "humidity"
- Addgene vectors should be appropriately acknowledged, including the depositing author, as specified on the addgene website.
- Primer sequences for site-directed mutagenesis and standard cloning should be included.
- Line 484 TMB is quenched with "0.2 M H₂SO₄" while on line 561 with "2 N sulfuric acid".
- Line 543 Should be DMEM media instead of DEME?
- Line 549 Elisa well plates should be specified, cat number?
- Line 56 "TMP" should be "TMB".

All the questions were addressed in the revised manuscript. The corrections were highlighted with red letters.

Comments on the Supplementary Information

Supp. Figure 6. Ladder should be labelled on the gel.

Ladder was labeled on the gel image.

Supp. Figure 8. It would be helpful to see a higher exposure of the Concanavalin-A blot.

Supplementary Fig. 8 was revised to address the questions raised by all reviewers.

REVIEWERS' COMMENTS:

Reviewer #1 (Remarks to the Author):

In the present manuscript by Lee et. al., have suggested generation of FC-aglycosylated antibody-lectin coupled immunoassay for the quantification of tumor markers (ALIQUAT) tool; I could see this tool is of clear interest. In the revised version of the manuscript, I could see some improvement in the quality. I have a couple of suggestions.

1. Supplementary figure 8B. The authors have used right approach and the experiment/tool is working. Unfortunately, control cases (Glyco-AB1/2 1E5) were overloaded as compared to test case (eg. 1E5). The result of the experiment is not informative as presented. The outcome of the result would be meaningful if all the samples loaded equally, control antibodies are overloaded in the experiment, and blotted for IgG and ConA as was done (both low and high exposure would be ideal).
2. Regarding the inhibition assay (whatever be the outcome of the experiment), it would be meaningful in my view to include the data in the supplementary. My understanding is that the authors have done the inhibition experiment to reach to the conclusion that Lfucose does not inhibit PhoSL binding in their assay. It is important to note that PhoSL is shown to bind core α 1-6-fucosylated N-glycans. Also, PhoSL is shown to bind fucose as defined in the NMR analysis (Yamasaki et al, 2018).
3. Line 28-29: The author state that because antibodies show cross-reactivity with lectin due to N-glycans in Fc region. This statement is simply not correct because N-glycans as well as even O-glycans on other parts of the antibody can potentially show some level of cross reactivity with lectin/s unless the authors specify the particular lectin/s. The authors may consider deleting that statement in this context.
4. Line 82: Authors should consider using -To resolve these issues, we genome-engineered mice by editing N ... Instead of- To resolve these problems straight forward, we genome engineered mice by editing N...
5. Please revisit the title of the supplementary figure 9: Confirmation of aglycosylation of monoclonal antibodies by MS/MS analysis. It appears to me that is an error should be aglycosylation of AFP-L3

Reference:

Yamasaki et. al. The trimeric solution structure and fucose-binding mechanism of the core fucosylation-specific lectin PhoSL, Scientific Reports 8, Article number: 7740 (2018)

Reviewer #2 (Remarks to the Author):

My concerns have been fully addressed and I would recommend acceptance of this MS to Communications Biology.

Reviewer #3 (Remarks to the Author):

The authors have appropriately responded to the queries and solved the previous issues. This is nice work that should be of interest to the readership of Communications Biology.

Responses to reviewer comments:

Reviewer #1 (Remarks to the Author):

1. Supplementary figure 8B. The authors have used right approach and the experiment/tool is working. Unfortunately, control cases (Glyco-AB_{1/2} 1E5) were overloaded as compared to test case (eg. 1E5). The result of the experiment is not informative as presented. The outcome of the result would be meaningful if all the samples loaded equally, control antibodies are overloaded in the experiment, and blotted for IgG and ConA as was done (both low and high exposure would be ideal).

The Western blot analysis was renewed and provided in Supplementary Figure 8B.

2. Regarding the inhibition assay (whatever be the outcome of the experiment), it would be meaningful in my view to include the data in the supplementary. My understanding is that the authors have done the inhibition experiment to reach to the conclusion that Lfucose does not inhibit PhoSL binding in their assay. It is important to note that PhoSL is shown to bind core α 1–6-fucosylated N-glycans. Also, PhoSL is shown to bind fucose as defined in the NMR analysis (Yamasaki et al, 2018).

We carefully re-examined the inhibition of AFP-L₃-lectin interactions by L-fucose and found that L-fucose, in fact, showed an inhibitory effect on the interactions, further confirming ALIQUAT method may provide a valid analytical platform for the assay of specific glycoforms. The previous failure to the inhibition assay is likely to originate from a problem during the purification of aglycosylated antibodies. Thank you for the comments on this important issues.

3. Line 28-29: The author state that because antibodies show cross-reactivity with lectin due to N-glycans in Fc region. This statement is simply not correct because N-glycans as well as even O-glycans on other parts of the antibody can potentially show some level of cross reactivity with lectin/s unless the authors specify the particular lectin/s. The authors may consider deleting that statement in this context.

The sentence was corrected to clear the confusions that the reviewer pointed.

4. Line 82: Authors should consider using- To resolve these issues, we genome-engineered mice by editing N ... Instead of- To resolve these problems straight forward, we genome engineered mice by editing N...

The sentence was corrected according to the reviewer's suggestion.

5. Please revisit the title of the supplementary figure 9: Confirmation of aglycosylation of monoclonal antibodies by MS/MS analysis. It appears to me that is an error should be aglycosylation of AFP-L₃.

The sentence was corrected according to the reviewer's suggestion.